# Widespread intronic polyadenylation diversifies immune cell transcriptomes

Irtisha Singh[1,2], Shih-Han Lee[3], Adam S. Sperling[4], Mehmet K. Samur[4], Yu-Tzu Tai[4], Mariateresa Fulciniti[4], Nikhil C. Munshi[4], Christine Mayr [3] & Christina S. Leslie[1]

Alternative cleavage and polyadenylation (ApA) is known to alter untranslated region (3′UTR) length but can also recognize intronic polyadenylation (IpA) signals to generate transcripts that lose part or all of the coding region. We analyzed 46 3′-seq and RNA-seq profiles from normal human tissues, primary immune cells, and multiple myeloma (MM) samples and created an atlas of 4927 high-confidence IpA events represented in these cell types. IpA isoforms are widely expressed in immune cells, differentially used during B-cell development or in different cellular environments, and can generate truncated proteins lacking C-terminal functional domains. This can mimic ectodomain shedding through loss of transmembrane domains or alter the binding specificity of proteins with DNA-binding or protein–protein interaction domains. MM cells display a striking loss of IpA isoforms expressed in plasma cells, associated with shorter progression-free survival and impacting key genes in MM biology and response to lenalidomide.

[1] Computational and Systems Biology Program, Memorial Sloan Kettering Cancer Center, New York, NY 10065, USA. [2] Tri-I Program in Computational Biology and Medicine, Weill Cornell Graduate College, New York, NY 10065, USA. [3] Cancer Biology and Genetics Program, Memorial Sloan Kettering Cancer Center, New York, NY 10065, USA. [4] Lebow Institute of Myeloma Therapeutics and Jerome Lipper Multiple Myeloma Center, Dana-Farber Cancer Institute, Harvard Medical School, Boston, MA 02215, USA. Correspondence and requests for materials should be addressed to C.M. (email: mayrc@mskcc.org) or to C.S.L. (email: cleslie@cbio.mskcc.org)

Alternative cleavage and polyadenylation (ApA) is generally viewed as the selection of ApA signals in the 3′ untranslated region (3′UTR), leading to the expression of different 3′UTR isoforms that code for the same protein. Recent computational analyses of 3′-end sequencing data have characterized the nature and extent of ApA in mammalian 3′UTRs[1–7]. For example, analysis of a human ApA tissue atlas established that half of human genes express multiple 3′UTRs, enabling tissue-specific post-transcriptional regulation of ubiquitously expressed genes[1]. However, ApA events can also occur in introns rather than 3′UTRs, generating either non-coding transcripts or transcripts with truncated coding regions that lead to loss of C-terminal domains in the protein product.

The most famous example of cell type-specific usage of an intronic polyadenylation (IpA) signal occurs in the immunoglobulin M heavy chain (*IGHM*) locus[8,9]. In mature B cells, recognition of the polyadenylation signal in the 3′UTR produces the full-length message, including two terminal exons that encode the transmembrane domain of the plasma membrane-bound form of immunoglobulin M (IgM; Fig. 1a). In plasma cells, usage of an IpA signal instead results in expression of an IpA isoform lacking these two terminal exons, leading to loss of the transmembrane domain and secretion of IgM antibody. Many additional IpA-generated truncated proteins have been described[10,11], including the soluble forms of epidermal growth factor and fibroblast growth factor receptors and a truncated version of the transcription factor NFI-B[12]. The IpA isoform of the interferon-induced anti-viral enzyme OAS1 generates an enzyme of comparable enzymatic activity as the full-length transcript but contains a hydrophobic C terminus rather than an acidic C terminus, suggesting that the two isoforms may interact with different cofactors or cellular structures[13]. Other examples include the transcription factor SREPF, whose IpA isoform can act as a developmental switch during spermatogenesis[14].

In the splicing literature, isoforms generated through recognition of an IpA signal are often described as 'alternative last exon' events[15]. Genes that generate IpA isoforms are thought to harbor competing splicing and polyadenylation signals, producing a full-length messenger RNA (mRNA) when splicing outcompetes polyadenylation and otherwise producing a truncated mRNA[16]. As the defining event is the recognition of an IpA signal, we call these transcripts IpA isoforms. It is now possible to recognize the widespread expression of IpA isoforms through the analysis of 3′-end sequencing data.

We identified robust ApA events that occur in introns and quantified IpA isoform expression using 3′-seq across human tissues, immune cells, and in multiple myeloma (MM) patient samples. We focused on immune cells because it is feasible to obtain pure populations of primary cells and because B cells expressed the largest number of IpA isoforms in our previous tissue atlas[1]. Through integration with RNA-seq profiles in B-lineage and MM cells as well as external data sets and annotation databases, we assembled an atlas of confident IpA isoforms supported either by independent data sources or very highly expressed in at least one cell type. We found that IpA isoforms are widely expressed, most prevalently in blood-derived immune cells, and that generation of IpA isoforms is regulated during B-cell development, between cellular environments, and in cancer. IpA events in immune cells are enriched at the start of the transcription unit, leading to IpA isoforms that retain none or little of the coding region (CDR) and hence represent a class of robustly expressed non-coding transcripts. IpA events that occur later in transcription units can lead to truncated proteins often lacking repeated C-terminal functional domains and thus contribute to the diversification of the proteome.

## Results

### 3′-seq analysis reveals widespread intronic polyadenylation.
To assemble an atlas of IpA isoforms, we used our previously published 3′-seq data set from normal human tissues (ovary, brain, breast, skeletal muscle, testis), cell types (embryonic stem (ES) cells, naive B cells from peripheral blood (blood NB)), and cell lines[1] and combined it with a newly generated data set from normal and malignant primary immune cells. The new immune cell profiles ($n = 29$) were all performed with biological replicates and included lymphoid tissue-derived naive B cells (NB), memory B cells (MemB), germinal center B cells (GCB) and CD5+ B cells (CD5+B), blood T cells and plasma cells (PCs), and MM derived from bone marrow aspirates (Supplementary Tables 1 and 2). We adapted our previously described computational pipeline to process 3′-seq libraries and detect and quantify ApA events, including intronic as well as 3′UTR events, while removing technical artifacts (see Methods)[1]. All subsequent analyses were restricted to protein coding genes. For additional evidence in support of IpA isoforms, we performed RNA-seq profiling in the same normal and malignant B cell types, where possible for the same samples (Supplementary Table 3).

We confirmed from both 3′-seq and RNA-seq data that the IpA isoform of *IGHM* is highly expressed in PCs while the full-length transcript, encoding membrane-bound IgM, is the dominant isoform in NB cells (Fig. 1b). Analysis of 3′-seq also revealed putative IpA isoforms, including in the locus of *GTF2H1*, encoding a subunit of general transcription factor II H, and *RAB10*, encoding a member of the Ras oncogene family of small GTPases (Fig. 1c). Like 3′UTR isoforms, IpA isoforms display differential expression across tissues and cell types. For example, the IpA isoform of *GTF2H1* is well expressed in skeletal muscle and immune cells, and indeed is the only isoform expressed in PC, blood NB, and T cells; these three cell types are also the only ones to express the IpA isoform of *RAB10*. To validate IpA events identified by 3′-seq, we used RNA-seq data from the same cell types to confirm intronic read coverage upstream but not downstream of the IpA event, as in PC RNA-seq coverage flanking the intronic 3′-seq peak in *GTF2H1* (Fig. 1d). Formally, we can test if RNA-seq read counts are significantly higher in intronic windows chosen upstream compared to downstream of IpA events (see Methods)[17]. We confirmed significantly differential coverage at 29% ($n = 1670$) of IpA events from our 3′-seq peak calls (false discovery rate (FDR)-adjusted $P < 0.1$) versus almost no significant differences at randomly chosen positions in introns (Supplementary Fig. 1a).

To assemble an atlas of confident IpA events, we compared each intronic 3′-seq peak against external annotation and data sources (see Methods, Supplementary Fig. 1b,c). Briefly, IpA events that overlapped with the last exon of annotated isoforms in RefSeq, UCSC (University of California Santa Cruz), and Ensembl were first added to the atlas (2241 events); unannotated IpA events that satisfied the test for differential upstream vs. downstream RNA-seq coverage were added next (907 events); unannotated IpA events without differential RNA-seq coverage but supported in data sets from other 3′-end sequencing protocols were then included (1332 events)[18]. We next added IpA events that lacked the previous sources of evidence but had RNA-seq support of the cleavage event—i.e., reads overlapping untemplated adenosines in the polyA tail (124 events). Finally, events with high expression in at least one cell type were also included (323 events). 13% ($n = 743$) of IpA events could not be validated by any of these criteria and thus were excluded from further analysis (Supplementary Fig. 1c). Overall, the atlas contains 4927 confident IpA events in 3431 protein coding genes, 55% of which are unannotated in RefSeq, UCSC, and Ensembl (Supplementary Fig. 1c). Similar proportions of annotated and unannotated IpA

isoforms were validated by various kinds of supporting evidence (Fig. 1e). Although we used other 3′-end sequencing data sets as an evidence source, only 54% (2677/4927) of IpA atlas events appear in polyA_DB 3 (http://exon.umdnj.edu/polya_db/), a resource based on 3′READS data[18], and 72% (3581/4927) have support from existing PolyA-seq, as they do not include the

immune cell types that we profile here. Therefore, while our atlas does not exhaustively sample human tissues, we have assembled the most comprehensive IpA resource to date.

**IpA is most prevalent in circulating immune cells**. We determined the prevalence of IpA across normal tissues and cell types

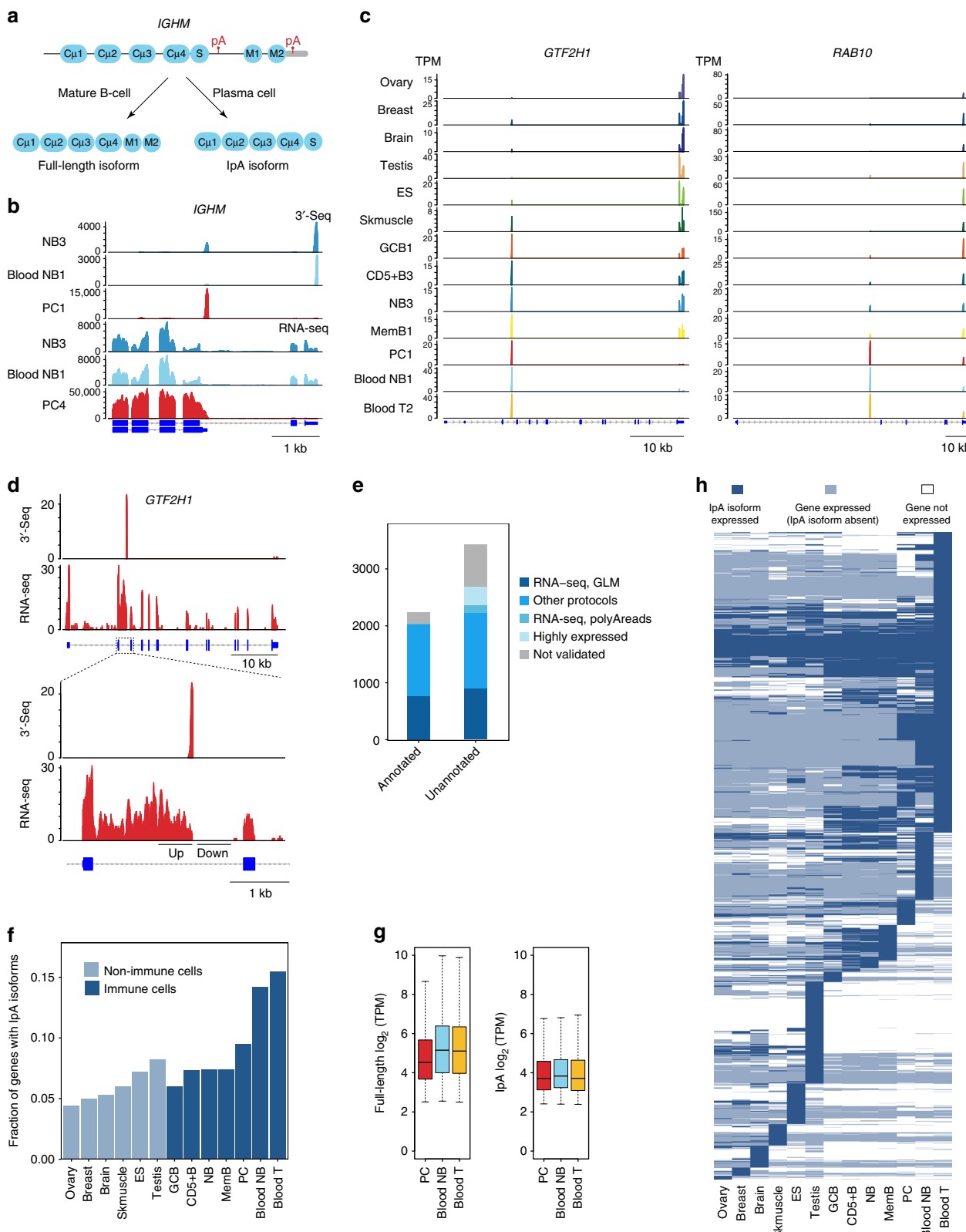

by computing the fraction of genes expressing at least one atlas IpA isoform out of all expressed genes in each cell type (Fig. 1f). Blood T cells had the highest fraction of genes with IpA isoforms (0.16) while ovary had the lowest fraction (0.04). In immune cells, 6–16% of genes expressed generate IpA isoforms compared to only 4–8% of genes in complex tissues, consistent with early IpA studies that used complementary DNA (cDNA) and EST (expressed sequence tag) data[19,20]. Notably, blood NB cells expressed 1114 IpA isoforms compared to only 721 IpA isoforms for tissue-derived NB cells, suggesting that the cellular environment has a strong effect on IpA isoform expression.

IpA isoforms are robustly expressed, with median expression of the same order of magnitude as for full-length isoforms (log$_2$ tags per million (TPM) of 3.71–3.83 for IpA isoforms in PCs, blood NB, and blood T versus log$_2$ TPM of 4.53–5.15 for full-length transcripts; Fig. 1g). Therefore, IpA isoforms are not 'transcriptional noise' produced from recognition of 'cryptic' sites, but rather represent major mRNA isoforms generated from alternative mRNA processing.

Figure 1h shows the tissue-specific expression of IpA isoforms, requiring a mean isoform expression level over 5 TPM across replicates to be considered 'expressed'. A majority of IpA isoforms with reproducible expression patterns are expressed in immune cell types ($n = 3365$), and almost all of these in at least two immune cell types. Non-immune tissues like testis and ES cells express tissue-specific IpA isoforms, but the majority are produced from tissue-specific genes.

**Cell types with frequent IpA express shorter isoforms.** To begin to assess the impact of IpA isoform expression, we computed the fraction of retained CDR for each IpA isoform relative to the full-length annotated CDR. The histogram of retained CDR fraction for atlas events showed a uniform distribution except for a substantial overrepresentation of IpA isoforms that lose all or almost all of the CDR (Fig. 2a). However, an examination of similar histograms across individual tissues and cell types revealed a more nuanced picture (Fig. 2b), where IpA events near the start of the transcription unit dominate in blood and bone marrow-derived immune cells, while brain and ES cells preferentially generate IpA events close to the end of transcription units. In testis and tissue-derived B cells, we found an intermediate pattern.

We observed a significant negative correlation across tissues between the frequency of IpA isoform expression and length of retained CDR ($r = -0.86$, Fig. 2c). Further, cell types with a tendency to produce longer 3′UTRs also prefer to use IpA events near the 3′ ends of transcription units ($r = 0.60$, Fig. 2d).

We use the term 5′IpA for IpA isoforms that retain less than 25% of the CDR and 3′IpA for the remainder. Both 5′IpA and 3′IpA events occur in introns that are significantly longer than the

introns from the same genes that contain no IpA events or from genes that only express full-length transcripts (one-sided Wilcoxon rank-sum test, $P < 10^{-20}$ for all three comparisons, Fig. 2e)[20]. Similarly, 5′IpA and 3′IpA isoforms are expressed from significantly longer transcription units than non-IpA genes (one-sided Wilcoxon rank-sum test, $P < 10^{-20}$ for both comparisons, Fig. 2f). 3′IpA atlas events have higher conservation by PhastCons in the sequence surrounding the polyadenylation signal compared to 5′IpA atlas events (one-sided Wilcoxon signed-rank test, $P < 10^{-66}$; Fig. 2g); however, 5′IpA events still show higher conservation than randomly chosen intronic polyadenylation signals with no 3′-seq coverage (one-sided Wilcoxon signed-rank test, $P < 10^{-68}$; Fig. 2g)[21].

An early study using cDNA and EST data[20] defined two types of IpA isoforms based on the structure of the terminal exon: 'composite terminal exon', where a donor splice site (5ss) is not recognized and the entire sequence from this donor splice site to the intronic cleavage site is included in the isoform; and 'skipped terminal exon', where the IpA isoform introduces a new small exon ending at the intronic cleavage site, requiring recognition of the previous donor splice site and a new acceptor splice site (Supplementary Fig. 2a). Through de novo assembly of RNA-seq data and comparison to 3′-seq in common cell types, we assembled the transcript structure for 2675 IpA atlas isoforms (see Methods). Of these, 1648 (61.6%) displayed loss of recognition of a donor splice site with 'composite' terminal exon, while the remainder introduced a new exon.

Interestingly, skipped terminal exon IpA isoforms predominantly use 5′IpA sites while composite terminal exon IpA sites occur throughout the transcription unit (Supplementary Fig. 2b). Skipped terminal exon IpA sites occur in much longer introns than those with composite terminal exon IpA sites or introns of non-IpA genes (one-sided Wilcoxon rank-sum test, $P < 10^{-20}$, Supplementary Fig. 2c)[20]. Both kinds of IpA events occurred in genes with longer transcription units than non-IpA genes (one-sided Wilcoxon rank-sum test, $P < 10^{-20}$, Supplementary Fig. 2c). Findings about the relative strengths of 5ss and 3ss signals were largely consistent with the earlier study (Supplementary Fig. 2d)[22]. Finally, 84% of skipped terminal exon IpA events and 80% of composite terminal exon IpA events are associated with AAUAAA/AUUAAA.

Previously, U1 small nuclear ribonucleoprotein (snRNP) expression and the presence of U1 snRNP motifs early in the transcription unit were found to play a crucial role in preventing premature cleavage and polyadenylation[23,24]. Consistent with these observations, we found that genes that express IpA isoforms contain a higher frequency of polyadenylation signals within their transcription unit and are depleted for U1 snRNP signals, as compared to genes that only express 3′UTR isoforms (Fig. 2h, i). To control for background AT content, we divided both IpA genes and non-IpA genes into those falling in high AT (AT

**Fig. 1** Widespread intronic polyadenylation in immune cells. **a** Schematic representation of full-length and IpA isoform of *IGHM* expressed in mature B cells and plasma cells (PC). **b** The 3′-seq (tags per million (TPM)) and RNA-seq (read coverage) tracks showing expression of the IpA and full-length mRNA isoforms of *IGHM* (ENSG00000211899), encoding the immunoglobulin mu heavy chain, IgM. The full-length isoform is expressed in NB from blood and lymphoid tissue and includes two exons encoding the C-terminal transmembrane domain of membrane-bound IgM. The IpA isoform is expressed in PCs obtained from bone marrow. It lacks the transmembrane domain which leads to expression of soluble IgM. **c** The 3′-seq tracks showing IpA isoform expression for two genes across human tissues and immune cell types. **d** RNA-seq coverage of intronic regions flanking IpA sites. A GLM-based test is used to validate the IpA isoforms. An isoform is considered validated if there is a significant difference (FDR-adjusted $P < 0.1$) in read counts in windows located up- and downstream of the putative IpA site. **e** The fraction of IpA isoforms validated by read evidence from independent data sets is shown for annotated and unannotated IpA isoforms. IpA isoforms present in RefSeq, UCSC genes, and Ensembl databases are considered to be annotated. **f** The fraction of expressed genes that generate IpA isoforms is shown for each cell type. **g** Expression levels (log$_2$ TPM) for full-length mRNAs and IpA isoforms are shown as boxplots for in PCs, blood NB, and T cells. IpA isoforms are robustly expressed as full-length mRNA expression is 4.53, 5.15, and 5.11, respectively, compared to 3.71, 3.83, and 3.71 for IpA isoforms. **h** Tissue-specific expression of IpA isoforms. Each row represents a gene. Dark blue, IpA isoform is expressed ($\geq$5 TPM); light blue, IpA isoform is not expressed, but full-length mRNA is expressed ($\geq$5.5 TPM); and white, gene is not expressed

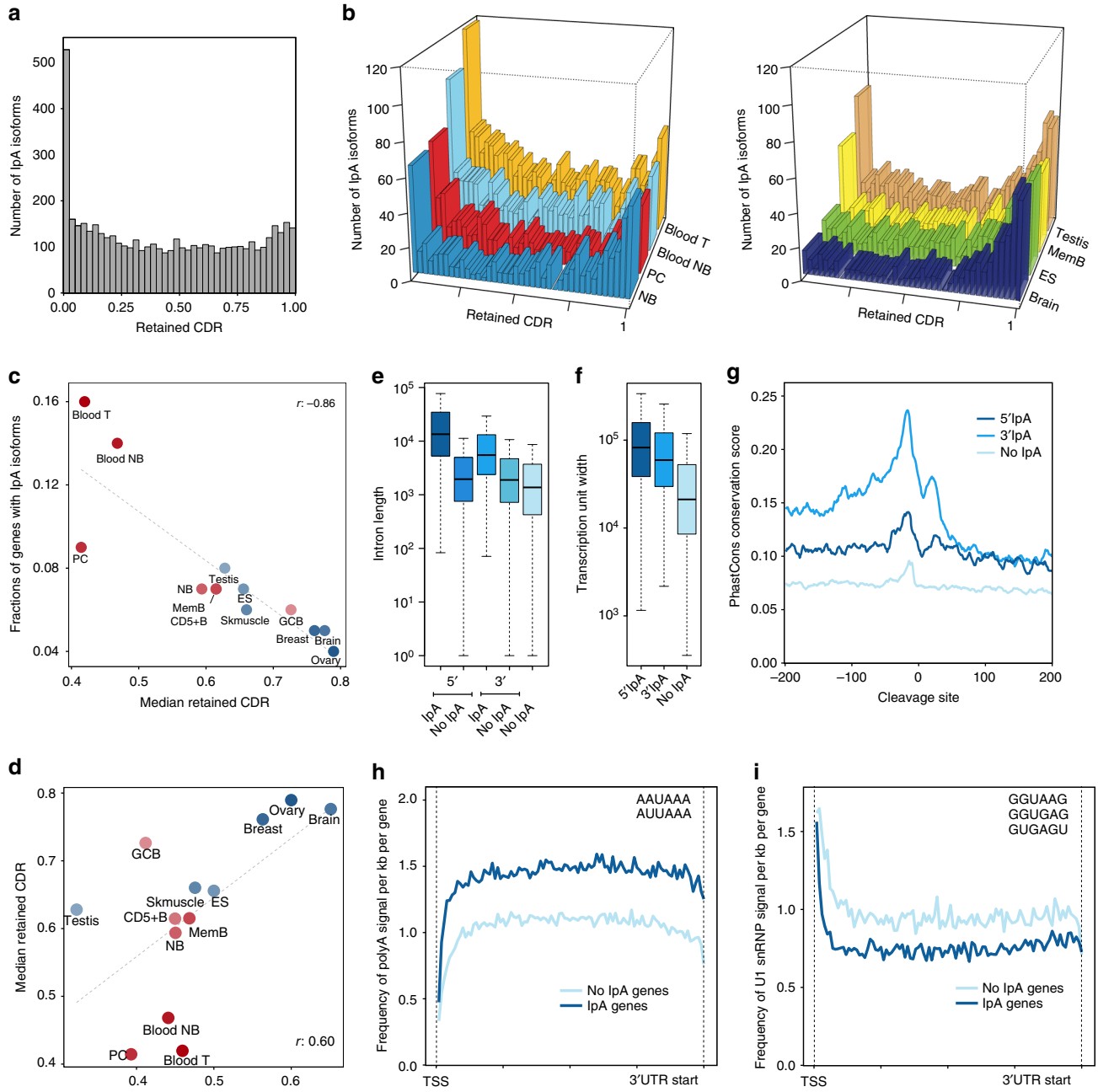

**Fig. 2** Enrichment of IpA sites at the start of transcription units. **a** The fraction of retained coding region (CDR) was calculated as the nucleotides from the start codon to the end of the exon located upstream of the IpA peak, divided by all coding nucleotides of the longest annotated open reading frame and is shown for all IpA isoforms in the atlas. **b** As in **a**, but shown for individual cell types. **c** Correlation between the median retained CDR with the fraction of genes that generate IpA isoforms in each sample (Pearson's correlation coefficient, $r = -0.86$). Tissues with a higher proportion of IpA isoforms generate IpA isoforms with shorter CDRs. **d** Correlation between the median retained CDR and the median usage of the distal ApA site in the 3′UTR (Pearson's correlation coefficient, $r = 0.60$). Tissues with shorter 3′UTRs have IpA isoforms with shorter CDRs. **e** IpA isoforms occur in long introns. The introns in which 5′IpA events occur are longer than the other introns of the same genes (one-sided Wilcoxon rank-sum test, $P < 10^{-20}$). Similarly, the introns in which 3′IpA events occur are longer than the remaining introns of those genes (one-sided Wilcoxon rank-sum test, $P < 10^{-20}$). If taken together, then the introns in which IpA events occur are longer than the introns of the genes that only express full-length isoform (one-sided Wilcoxon rank-sum test, $P < 10^{-20}$). **f** IpA isoforms occur in genes with long transcription units. Genes that express IpA isoforms have longer transcription units compared to genes that only express full-length isoforms (one-sided Wilcoxon rank-sum test, $P < 10^{-20}$). **g** Higher conservation around the cleavage sites of IpA isoforms. The plot shows PhastCons scores of 200 nt upstream and downstream of IpA cleavage sites ($x = 0$). 5′IpA and 3′IpA events both have significantly higher conservation flanking the cleavage site compared to corresponding regions of randomly selected polyA signals (AAUAAA) in introns lacking IpA events (one-sided Wilcoxon signed-rank test, $P < 10^{-68}$ for both comparisons). **h** Genes with IpA isoforms ($n = 3481$) are enriched for polyadenylation sites compared with genes that do no generate IpA isoforms ($n = 12,092$) (one-sided Wilcoxon signed-rank test, $P < 10^{-18}$). The frequency of polyadenylation sites was counted from the TSS to the beginning of the 3′UTR and is shown as the average number of signals per kb per gene. **i** As in **h**, but U1 binding sites are shown. IpA genes are depleted for U1 snRNP signals (one-sided Wilcoxon signed-rank test, $P < 10^{-18}$)

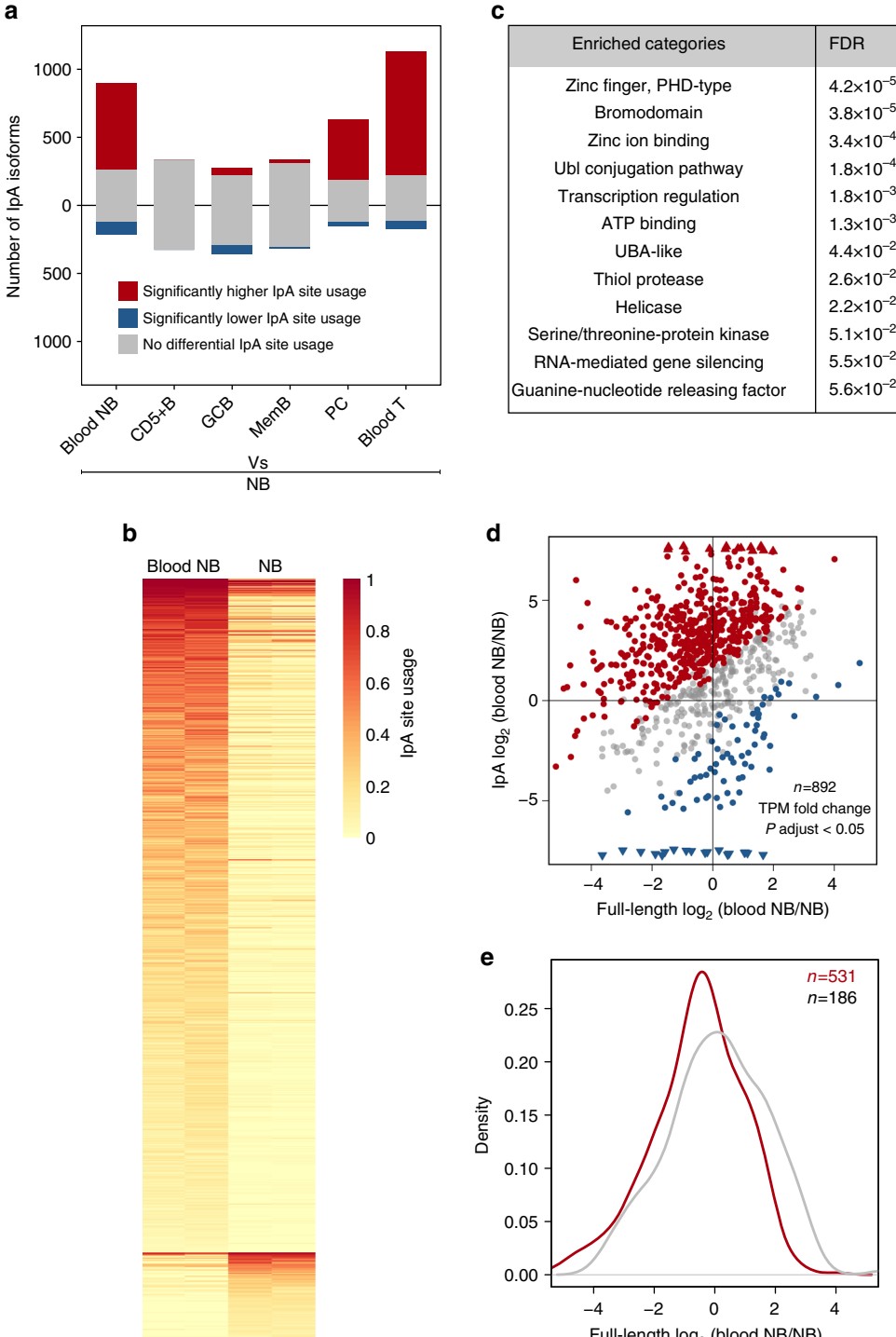

**Fig. 3** Dynamic expression of IpA isoforms in immune cells. **a** Number of IpA isoforms with differential usage of IpA sites between NB from lymphoid tissue versus other immune cells (FDR-adjusted $P < 0.05$). **b** Heatmap showing IpA site usage of IpA isoforms with significantly different usage (FDR-adjusted $P < 0.05$) between NB derived from blood or lymphoid tissue ($n = 720$). Each row indicates an IpA isoform. **c** Enrichment of gene ontology terms for the genes shown in **b**. **d** Fold change of IpA isoform and full-length mRNA expression in blood versus lymphoid tissue-derived NB by TPM. All the genes that were tested for differential usage are shown ($n = 892$). If a gene had multiple IpA isoforms, then the one with the most significant differential IpA usage is shown. IpA isoforms with significantly different usage (FDR-adjusted $P < 0.05$) are highlighted in red (higher usage) or blue (lower usage). **e** Significant downregulation of full-length mRNAs in genes with significant IpA isoform expression (one-sided KS test, $P < 10^{-5}$). Shown are genes highlighted in red from **d**

content > 50%) versus low AT (AT content < 50%) regions and repeated the polyadenylation and U1 signal analysis (Supplementary Fig. 2e). We again found significant enrichment of polyadenylation signals in IpA genes vs. non-IpA genes in both high and low AT regions as well as depletion of U1 signals (one-sided Wilcoxon signed-rank test, $P < 10^{-10}$ for all four comparisons), although the effect sizes were more modest. However, IpA genes have high AT content compared to the genes lacking IpA events, so AT content is enough to segregate IpA genes from non-IpA genes (Supplementary Fig. 2f, Wilcoxon rank-sum test, $P < 10^{-20}$). Therefore, genomic architecture and sequence composition may facilitate IpA isoform expression, but

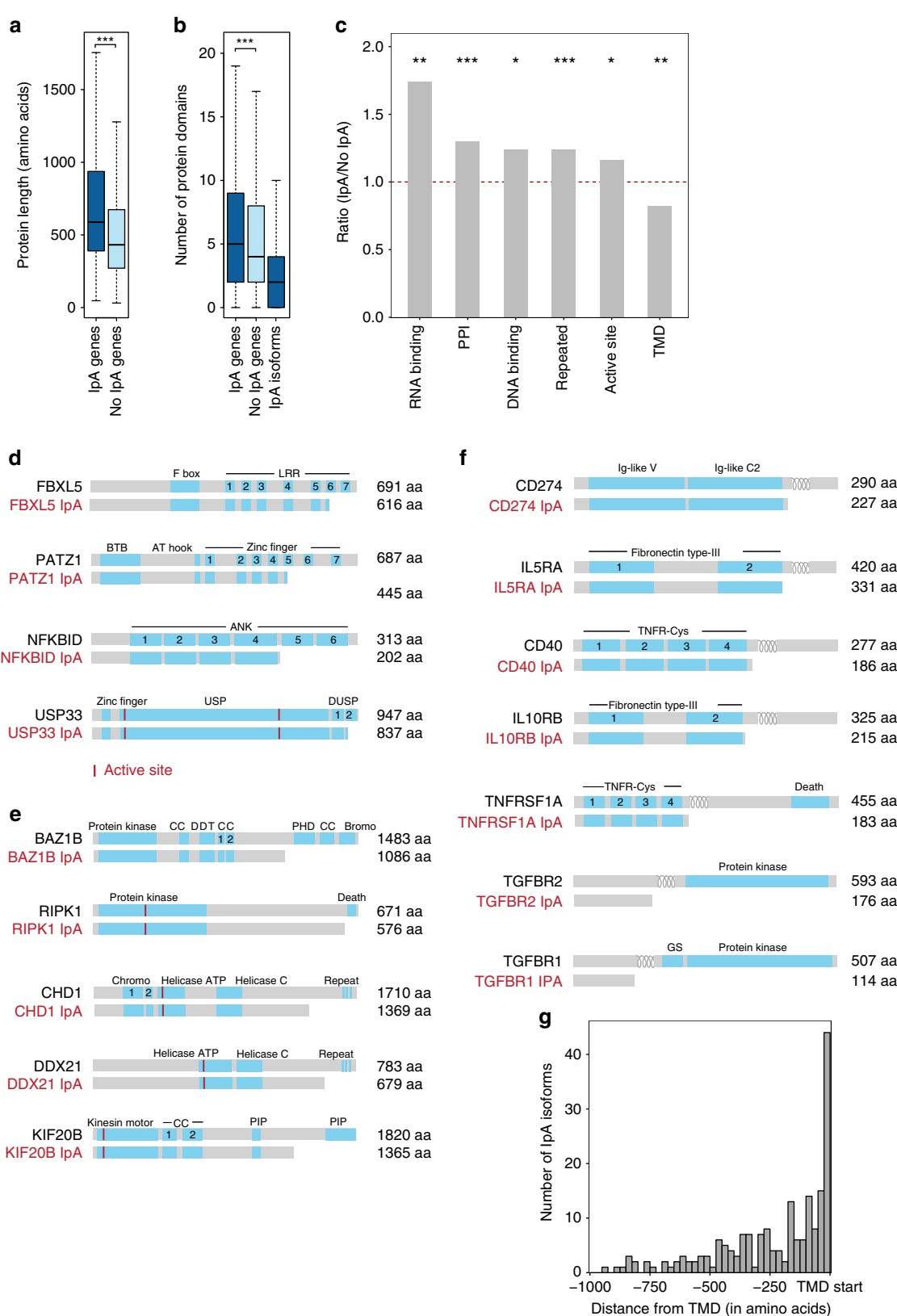

high background AT content is also strongly associated with IpA genes.

**IpA usage leads to modest full-length mRNA downregulation.** Next we used a generalized linear model (GLM) approach to determine significant changes in the relative expression of IpA isoforms compared to full-length transcripts (usage of IpA) across normal immune cells (see Methods)[1,17,25]. A majority of expressed IpA isoforms significantly differed in usage between NB cells from lymphoid tissue and blood T cells (950/1308, Fig. 3a, FDR-adjusted $P < 0.05$, Supplementary Data). We also found differential IpA usage between B-lineage cell types, with PCs in particular showing strikingly increased IpA site usage compared to tissue-derived NB cells (Fig. 3a). However, surprisingly, we found even more significant changes in IpA usage between NB cells from lymphoid tissue and blood (720/1113, Supplementary Data) than between different B-cell types (Fig. 3a, b). Thus, IpA isoform expression is not only cell-type differential but also changes between different cellular environments. Genes with differential usage of IpA isoforms between immune cell types were most strongly enriched for annotations including zinc-finger domains, bromodomains, and the ubiquitin-like conjugation pathway (Fig. 3c)[26].

Figure 3d plots the expression change of the IpA isoform against that of the full-length transcript and shows IpA genes that differentially increase (red points) or decrease (blue points) usage of their IpA isoforms in blood-derived compared with tissue-derived NB cells. Genes that increase IpA isoform usage in blood- versus tissue-derived NB cells significantly reduce expression of their full-length transcripts compared to genes without significant change in IpA usage (Fig. 3e, one-sided Kolmogorov–Smirnov (KS) test, $P < 10^{-5}$). However, the decrease in full-length isoform expression was modest, indicating that IpA usage does not predominantly result in a 'switch-like' change between full-length and IpA isoform expression.

**IpA diversifies the transcriptome by C-terminal domain loss.** We next observed that IpA genes encode full-length proteins that are significantly larger and contain more domains than non-IpA genes (Fig. 4a, median number of amino acids 588 vs. 432; Fig. 4b, median 5 vs. 4 domains). Notably, most IpA-generated truncated proteins still retain functional protein domains, suggesting that IpA helps diversify the transcriptome (Fig. 4b, median 2 domains). IpA genes preferentially encode proteins with RNA- or DNA-binding or protein–protein interaction (PPI) domains but avoid membrane proteins. Proteins encoded by IpA genes are also enriched in repeated domains (Supplementary Fig. 3a, Fig. 4c), which in a majority of cases are partially lost through IpA. For example, the full-length protein encoded by *NFKBID* has six ankyrin domains, while the IpA-generated truncated protein retains four of them. Similarly, the full-length protein of the transcription factor PATZ1 has seven zinc-finger domains, while different IpA isoforms are predicted to encode either four or five zinc-fingers (Fig. 4d). The partial loss of DNA-binding domains potentially changes DNA-binding specificity and therefore the set

of regulated target genes. Similarly, the partial loss of PPIs can change the binding affinity to protein interaction partners.

Among genes with a single IpA event and whose IpA isoform retains at least one protein domain ($n = 1405$), IpA results in a preferential loss of DNA-binding or PPI domains but avoids the loss of active sites (Supplementary Fig. 3b,c and see Methods). Loss of an active site, where substrate binding and catalysis take place, would make an enzyme dysfunctional, but IpA appears to avoid this outcome. IpA genes encode diverse proteins with enzymatic functions, including protein kinases, DNA or RNA helicases, or motor proteins (Fig. 4e). The IpA-generated truncated proteins retain their active sites but lose PPI domains, which may enable the enzymes to participate in different protein complexes or change the substrate. For example, the full-length protein kinase RIPK1 contains a C-terminal death domain that is excluded in RIPK1 IpA (Fig. 4e). BAZ1B, also called WSTF, is a multi-functional protein that contains an N-terminal protein kinase domain but can exclude C-terminal located coiled-coil, zinc-finger, and bromodomains. Also, helicases, including DDX21, DDX49, and DHX15, as well as motor proteins such as KIF20B retain their enzymatic function but generate proteins lacking interaction domains.

Membrane proteins are characterized by the presence of transmembrane domains (TMDs) and are significantly depleted among IpA genes (Fig. 4c, Supplementary Fig. 3a). Nevertheless, we found 673 IpA isoforms from 499 genes that encode transmembrane proteins and retain at least one protein domain. Among them, 207 IpA isoforms from 152 genes completely retained their TMDs, whereas 220 IpA isoforms from 175 genes lost their TMDs. Interestingly, IpA isoforms that retain the TMDs often encode intracellular membrane proteins that localize to mitochondria. In contrast, IpA isoforms that lose their TMDs are significantly enriched in signal peptides that are predominantly present in plasma membrane proteins (FDR, $P < 9.1 \times 10^{-29}$, Fig. 4f, see Methods). Many of them encode cytokine receptors, integrins, or growth factor receptors. Notably, regardless of the position of the TMD, the truncated protein generated by IpA usually terminates immediately before the TMD (Fig. 4g). As all of these candidates contain signal peptides at the N terminus, the IpA isoform produces a secreted form of the cytokine or growth factor receptor.

**5′IpA can produce robustly expressed non-coding RNAs.** A large fraction of IpA isoforms that are differentially used among normal immune cell types are in fact 5′IpA isoforms (487 out of 1281). Through de novo RNA-seq assembly, we were able to resolve the transcript structure for 954 of the 5′IpA isoforms (see Methods) and found that 469 of these have low predicted coding potential with open reading frames (ORFs) encoding fewer than 100 amino acids[27]. Therefore, they likely generate micropeptides or represent non-coding RNAs (Fig. 5, Supplementary Fig. 4)[28–31]. To assess potential functional consequences of these non-coding transcripts, we examined if RNA-binding proteins may preferentially bind to the exonized intronic sequences upstream of the IpA cleavage site. As shown in Fig. 5, the new exons are

**Fig. 4** IpA isoforms diversify the proteome. **a** Genes that express IpA isoforms encode significantly larger proteins compared to genes that only express full-length mRNAs (one-sided Wilcoxon rank-sum test, $P < 10^{-118}$); *$P < 0.01$; **$P < 10^{-5}$; ***$P < 10^{-10}$. **b** Genes that express IpA isoforms encode proteins with significantly more protein domains than genes that only express full-length mRNAs (one-sided Wilcoxon rank-sum test, $P < 10^{-14}$). IpA isoforms retain a median of two domains. **c** Genes that express IpA isoforms are enriched in proteins encoding RNA- and DNA-binding, PPI repeated domains and active sites compared to genes that only express full-length mRNAs. However, IpA genes are depleted for proteins encoding transmembrane domains (TMDs). **d** Protein models of full-length and IpA-generated truncated proteins are shown in gray for examples that contain repeated domains. Known protein domains are shown as blue boxes and repeated domains are numbered. **e** As in **d**, but shown for enzymes that retain their active sites but lose PPI domains. **f** As in **d**, but shown for plasma membrane proteins. The TMD is indicated by the loops. **g** Distance between the IpA event and the start of the TMD in IpA isoforms that completely lose their TMDs ($n = 272$)

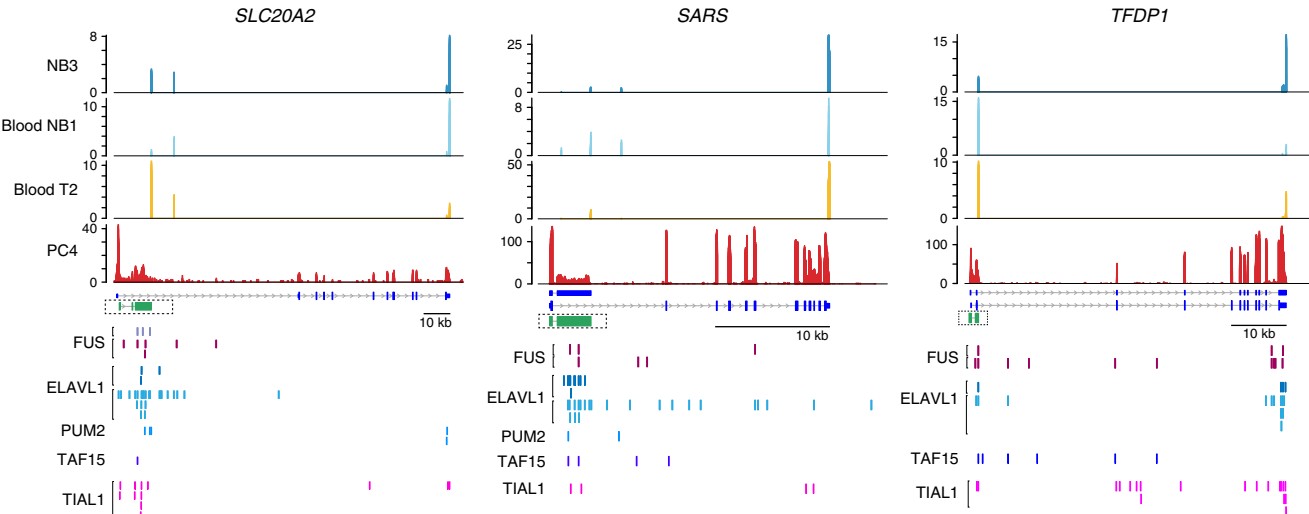

**Fig. 5** 5′IpA isoforms potentially express non-coding RNAs. Examples of 5′IpA isoforms are shown as in Fig. 1b. Also shown is the structure of the assembled IpA isoform transcripts in green. Enrichment of CLIP-seq tags over exonized introns of IpA isoforms are shown for the RNA-binding proteins FUS, ELAVL1, PUM2, TAF15, and TIAL1

enriched for cross-linking immunoprecipitation (CLIP)-seq peaks for RNA-binding proteins such as FUS, ELAVL1, PUM2, TAF15, and TIAL1 (binomial $Z > 10$, see Methods), which are typically enriched in the 3′UTRs of coding transcripts, but not for RNA-binding proteins usually bound to introns, supporting the exonic nature of the predicted non-coding transcripts.

**Multiple myeloma displays a widespread loss of plasma cell IpA isoforms.** As alternative 3′UTR isoform expression can be altered in cancer cells[6,32,33], we investigated whether IpA is also dysregulated in cancer. Since PCs express the highest number of IpA isoforms among the tissue-derived B cells, we compared IpA isoform expression between normal and malignant PCs, derived from MM patients ($n = 15$). As MM is a heterogeneous disease, we used hierarchical clustering based on IpA isoform expression to define three patient subgroups (Supplementary Fig. 5a and Supplementary Table 2). We then performed GLM modeling as described above to determine the differential relative expression of IpA isoforms versus full-length isoforms for each MM group compared to normal PCs. Whereas one patient group had an IpA profile comparable to normal PCs, two MM patient groups showed widespread loss of usage of PC IpA events (groups 1 and 2, Fig. 6a). We found that 44% of all PC-expressed IpA isoforms (480/1088, Supplementary Data) are lost in at least one patient group, while only 15 IpA sites show increased usage (FDR-adjusted $P < 0.05$). The significant events in patient group 1 largely represent a superset of those in group 2 (Fig. 5b).

Loss of IpA isoform expression in patient group 1 resulted in a significant increase of full-length mRNA expression (Fig. 6b, c). Genes with differential IpA usage in MM versus PC were again enriched for annotations such as bromodomain, transcriptional regulation, and ubiquitin-like conjugation pathway. In the majority of patient samples profiled (11 out of 15), the MM transcriptome is characterized by the loss of 480 IpA isoforms that are normally expressed in PCs. This is in contrast to 3′UTR regulation, where we found shortening of 3′UTRs in 126 and lengthening of 3′UTRs in 215 multi-UTR genes (MM group 1).

To investigate whether IpA isoform expression is correlated with any clinical factor, we examined a cohort of 286 MM patients with RNA-seq profiles and clinical information from a separate study[34]. Before proceeding, we established a set of IpA isoforms whose expression from 3′-seq is robustly represented by RNA-seq signal in the original 12 MM patients (see Methods). While RNA-seq signal is often not an accurate readout of IpA isoform expression levels (Supplementary Fig. 6a), we selected IpA isoforms with high correlation between RNA-seq and 3′-seq (Pearson's $r > 0.75$, $n = 28$) to use as a signature for the extent of IpA loss in the larger RNA-seq cohort.

Hierarchical clustering with respect to the signature IpA events segregated the 286 patients largely into two groups (see Methods, Supplementary Fig. 6b,c), group A ($n = 126$) with low median IpA usage and group B ($n = 160$) with high median IpA. Interestingly, group B patients, whose IpA usage is more similar to normal PCs, have improved progression-free survival ($P < 0.05$) compared to group A patients (Supplementary Fig. 6d). Since we noticed some heterogeneity in the groups, we applied a $k$-nearest-neighbor filter (see Methods) to remove patients whose neighbors had inconsistent cluster assignment. This improved the contrast between low IpA group A-filtered ($n = 64$) and high IpA group B-filtered patients ($n = 100$, Supplementary Fig. 6e) and difference in progression-free survival ($P < 0.028$, Fig. 6d). This analysis suggests either that IpA isoform loss is associated with faster progression or represents a more advanced disease state compared to MM cells that still express PC IpA isoforms.

Interestingly, one of the genes that displays loss of IpA isoform expression is the transcription factor *IKZF1*, a key gene in MM biology and the target of Cereblon-mediated degradation induced by lenalidomide, a MM therapeutic derived from thalidomide (Fig. 6e)[35,36]. The IpA isoform of *IKZF1* loses all zinc-finger domains encoded by the full-length transcript, potentially leading to expression of a truncated protein isoform with no known domain. While the IpA isoform is the dominant isoform in PC, it is almost completely lost in MM group 1 patients, which instead aberrantly express the full-length transcript. The gene *CUL4A* encodes a component of the DDB1-CUL4A-ROC1 E3 ubiquitin ligase complex involved in Cereblon (CRBN)-mediated degradation of IKZF1 by lenalidomide[35]. The IpA isoform of *CUL4A* is translated into a truncated protein (1–174 amino acids) that retains only its N-terminal domain (Fig. 6e). *CUL4A* is overexpressed/amplified in other cancers[37–40] and restricts cellular DNA damage repair[41], and sensitivity to thalidomide correlates with CUL4A expression in prostate cancer cell lines[42]. Similarly, the gene encoding

*IQGAP1*, a GTPase-activating scaffold protein involved in cell proliferation in MM, largely loses IpA isoform expression in MM[43]. This isoform lacks its Ras-GTP domain as well as most functional domains and is either non-coding or at best produces a truncated protein with only a fraction of the N-

terminal actin-binding domain (Fig. 6e). We validated the relative expression of *IKZF1*, *CUL4A*, and *IQGAP1* IpA isoforms at the mRNA level in our cohort of 15 MM patient samples and normal PCs by quantitative reverse transcription polymerase chain reaction (qRT-PCR; Fig. 6f). The ratio of IpA

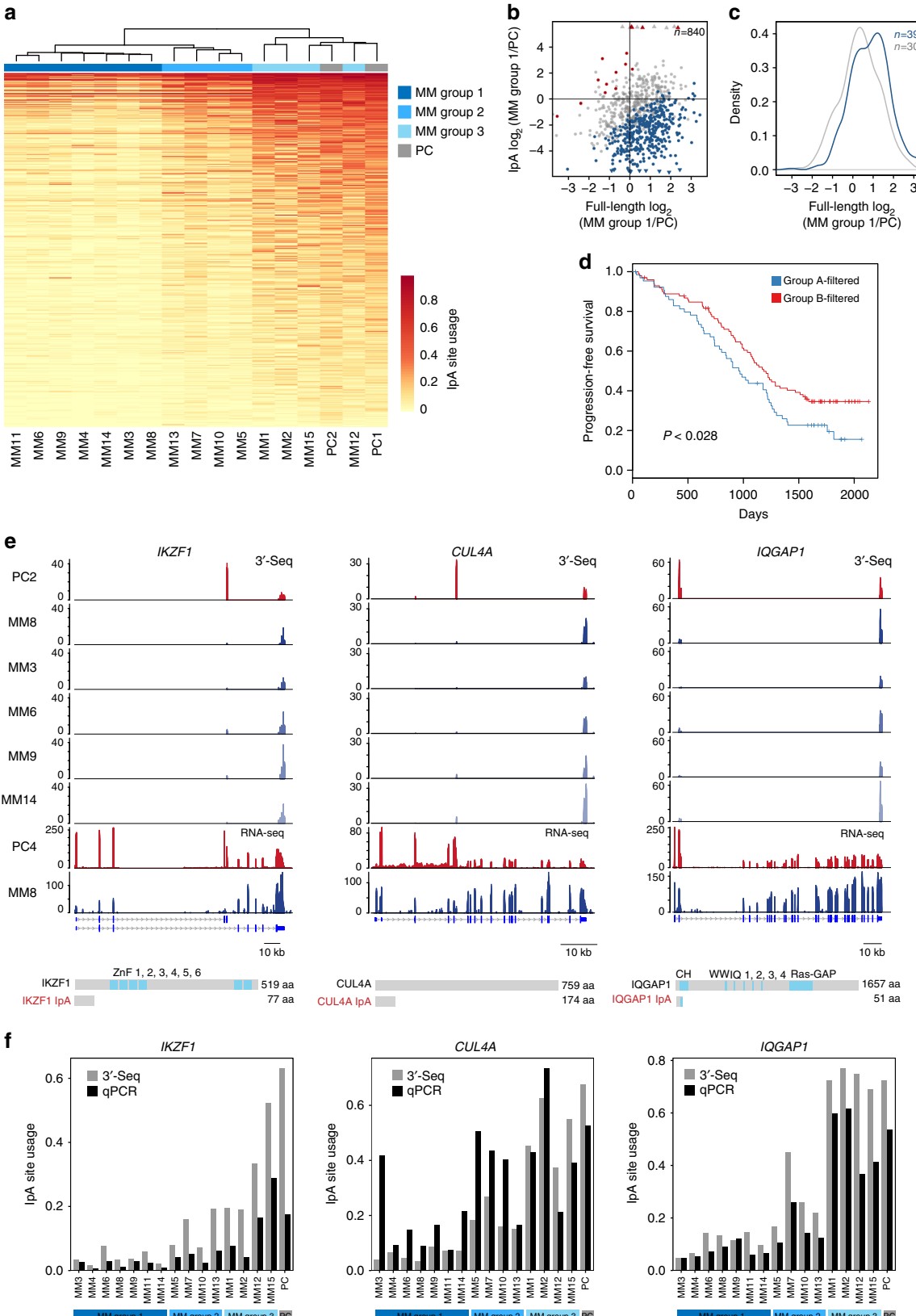

usage by qRT-PCR is concordant with 3′-seq, confirming the loss of IpA isoforms in a group of MM patients.

The expression of IpA isoforms of *IKZF1* and *CUL4A* may have important implications for the response of normal PCs and some MM patients to lenalidomide treatment. The Cereblon degron sequence is in a zinc-finger domain of IKZF1 (Supplementary Fig. 7a), and therefore the IpA isoform that is predominantly expressed in normal PCs cannot be targeted by lenalidomide. In fact, while lenalidomide effectively depletes malignant PCs, it is known to have little effect on normal PCs. The DDB1-CUL4A-ROC1 crystal structure shows that the N-terminal region of CUL4A interacts with DDB1 while the C-terminal region interacts with ROC1, which recruits the E2 ubiquitin-conjugating enzyme (Supplementary Fig. 7b, top)[44]. Furthermore, a truncated CUL4A (residues 1–297) has been shown to act in a dominant negative manner[45]. The *CUL4A*-IpA isoform retains its N-terminal domain and hence the potential to interact with DDB1 (Supplementary Fig. 7b, bottom). Based on these observations, we hypothesize that CUL4A-IpA may have the potential to act in a dominant negative manner in patients who express the CUL4A-IpA isoform (group 3 from Fig. 6a).

## Discussion

IpA isoform expression has previously been viewed as a form of alternative splicing involving alternative last exon usage[15,46], but the defining event is usage of an intronic alternative poly-adenylation signal. We performed comprehensive IpA analyses using 46 3′-seq and RNA-seq samples and identified 4927 high-confidence IpA events, the majority of them unannotated. We found that IpA is unexpectedly widespread and especially common among normal human immune cells. Expression of IpA isoforms is often robust—they do not represent 'cryptic' events or 'transcriptional noise'—and is regulated across normal cells and dysregulated in cancer.

Recently, individual examples of IpA isoforms have been studied in detail, including a truncated protein arising from IpA in the gene encoding platelet-derived growth factor receptor-α[47] and a non-coding IpA isoform transcribed from the helicase gene *ASCC3*[48]. However, the widespread nature of IpA isoform expression has escaped attention thus far, as RNA-seq analysis alone is unable to accurately identify mRNA 3′ ends. By combining 3′-seq and RNA-seq analyses, we identified and resolved the transcript structure of hundreds of new non-coding RNAs as well as truncated mRNAs predicted to generate proteins with alternative C termini. The IpA-generated truncated mRNAs are not subject to degradation by nonsense mediated decay, since their stop codons are followed by conventional mRNA 3′ ends and hence not premature.

IpA isoform expression is thought to be regulated by competition between splicing and cleavage-polyadenylation reactions[10,16]. Consistent with this model, we found that IpA genes have distinct structural and sequence properties that may predispose them toward IpA recognition. IpA genes have longer introns, longer transcription units, and higher AT content; after

correcting for AT content, IpA genes retain an enrichment of polyadenylation signals and a depletion of U1 snRNP signals relative to non-IpA genes[23,24]. Nevertheless, the tissue-specific differential expression of many IpA isoforms also suggests more complex regulation of production or stability. It is possible that degradation factors, including components of the RNA exosome, are downregulated in immune cells, leading to more frequent IpA[49]. Alternatively, there may be differential expression of splicing factors, such as heterogeneous nuclear RNP (hnRNP) C and U2AF65, involved in the regulation of Alu exonization, a mechanism known to control the expression of intronic exons[50,51]. The fact that more prevalent use of IpA signals, shorter IpA isoforms, and shorter 3′UTRs are all correlated across tissues suggests that the abundance of the same global co-transcriptional factors may be partially responsible for all three properties.

The finding that long introns and long transcription units are more susceptible to IpA suggests that the processivity of the co-transcriptional machinery may also play a role in IpA expression. Intron retention is a prevalent feature of blood cell transcriptomes[52,53]. The tendency to retain certain introns may provide the polyadenylation machinery time to recognize IpA signals for cleavage and 3′ end processing. Interestingly, we did find a statistically significant co-occurrence of introns with IpA events and retained introns (Supplementary Fig. 8a). Prevalence of intron retention correlates with prevalence of IpA across cell types (Supplementary Fig. 8b) and retained introns are enriched for IpA in each cell type examined (Supplementary Fig. 8c). However, IpA events in introns with no evidence of intron retention have higher IpA site usage than those in retained introns (Supplementary Fig. 8d). Therefore, it is unclear from our data whether intron retention is a necessary mRNA processing step prior to IpA recognition and 3′ end formation, or whether IpA recognition can occur independently of intron retention.

A surprising finding was the enrichment of IpA isoforms at the 5′ end of the transcription unit in immune cells. We identified 469 5′IpA isoforms with predicted coding sequence generating fewer than 100 amino acids. These IpA isoforms are either non-coding or represent a source of micropeptides[28–31]. The cellular function of non-coding RNAs generated through IpA is unclear. There are reports of promoter-associated RNAs that initiate upstream of transcription start sites and regulate transcript expression through RNA interference or interaction with epigenetic modifying enzymes[54–57]. However, in matching RNA-seq data, we did not find read evidence upstream of transcription start sites associated with our predicted non-coding RNAs. Additionally, CLIP-sequencing data analysis showed that the exonized intronic sequence of the 5′IpA isoforms contains binding sites for RNA-binding proteins (RBPs); potentially, these non-coding RNAs serve as scaffolds for RBPs and thereby exert a regulatory role in *trans* on other RNAs.

The majority of IpA isoforms ($n = 2667$), however, are predicted to generate truncated proteins that retain at least one domain and have the potential to be functional. Notably, IpA genes encode larger proteins that contain significantly more

**Fig. 6** Loss of usage of IpA sites in MM. **a** Heatmap shown as in Fig. 3b, but for PCs and MM patient samples. MM samples were grouped according to IpA site usage and color-coded, and the IpA isoforms with significantly different usage compared to PCs are shown (FDR-adjusted $P < 0.05$, lower usage of IpA sites in MM, $n = 480$; higher usage of IpA sites in MM, $n = 15$, not shown). **b** As in Fig. 3d, but for MM group 1 versus PCs. Full-length and IpA isoform expression is shown and significantly different IpA isoforms are color-coded (FDR-adjusted $P < 0.05$). **c** As in Fig. 3b. Shown is a significant upregulation of full-length mRNA isoform expression (one-sided KS test, $P < 10^{-8}$) of genes highlighted in blue in (**b**). **d** Independent set of patients with RNA-seq expression ($n = 286$) were classified into two groups based on the IpA usage of a gene signature (see Methods). Patients in group A-filtered ($n = 64$) exhibit loss of recognition of IpA sites, while patients in group B-filtered ($n = 100$) exhibit higher usage of IpA sites like the PCs. Group B-filtered patients have significantly improved ($P < 0.028$) progression-free survival than group A patients. **e** Examples of IpA isoforms expressed in PCs, but significantly decreased in MM samples. Shown as in Fig. 1b. **f** Validation of expression levels of IpA isoforms from **e** in terms of usage by qRT-PCR

domains than proteins generated from non-IpA genes. Strikingly, proteins encoded by IpA genes are enriched for repeated protein domains, which in a majority of cases are only partially lost, thus modulating but not abolishing overall protein function. Moreover, the active sites of enzymes are generally retained in IpA-generated truncated proteins, again resulting in proteins with similar function as the full-length proteins but different affinity or different binding partners. Thus, the cell type-specific expression of truncated proteins generated through IpA may be a widely used mechanism to diversify the proteome, not a peculiarity of a few well-known examples like IgM.

Although IpA genes avoid membrane proteins overall, IpA can mimic ectodomain shedding in transmembrane proteins. Metalloproteinases such as ADAM10 or ADAM17 are known to release the ectodomains of several surface receptors, including tumor necrosis factor-α (TNFα), L-selectin, transforming growth factor-α, or CD40[58,59]. In several cases, the soluble ligands act as agonists or antagonists of the membrane-bound ligands; for example, membrane-bound Fas ligand kills T lymphocytes, while soluble Fas ligand blocks this activity[60]. In the vast majority of known cases, proteolytic cleavage occurs close to the plasma membrane, cutting at a site near the TMD to release the extracellular domain of the growth factor receptor or cytokine. Intriguingly, we found that IpA is another potential mechanism to produce soluble versions of membrane-bound receptors, as IpA-generated truncations also occur close to the TMD. This suggests that developmental regulation of membrane-bound versus secreted molecules—first described for IgM—is widespread and can mimic proteolytic cleavage. For example, IpA can generate soluble TNF receptor 1, which has been shown to block TNF activity and is associated with multiple sclerosis[61].

It is interesting to speculate why immune cells generate more IpA isoforms than solid tissues. One possibility is that circulating immune cells are specialized for secretion of cytokines, and IpA provides a mechanism for generation of soluble isoforms of proteins whose full-length isoforms are membrane bound. In addition, the immune system requires repeated execution of complex differentiation programs. Perhaps immune cells have acquired IpA events for specialization of transcriptional and post-transcriptional programs during cellular differentiation through partial loss of repeated DNA-binding and RBP domains. An important example is *IKZF1*, the gene encoding IKAROS, a transcription factor whose full-length form is predominantly expressed in mature B cells. The activity of IKZF1 is known to be essential for the development of B-cell precursors[62], but its DNA-binding activity appears to be reduced in PCs through a switch to expression of the apparently non-DNA-binding IpA isoform.

Indeed, IpA isoform expression changes in different stages of B-cell development and after environmental changes and importantly is dysregulated in cancer. A majority of the MM patients that we profiled showed a striking loss of IpA isoforms normally expressed in PCs. As a group, genes that lose IpA expression in MM samples compared to PCs also upregulate full-length transcript expression, presumably rescuing the function of the full-length protein to varying degrees. Here, *IZKF1* displays a switch-like loss of IpA isoform expression and rescue of full-length transcript expression in MM, providing a key therapeutic target through Cereblon-mediated degradation by lenalidomide[35]. Intriguingly, some MM patients retain expression of PC IpA isoforms that may be therapeutically relevant, such as *CUL4A*-IpA. CUL4A acts in the DDB1-CUL4A-ROC1 E3 ubiquitin ligase complex that interacts with Cereblon to ubiquitinate and degrade IKZF1. CUL4A-IpA may act as a dominant negative, able to bind DDB1 but unable to recruit the E2 enzyme required for substrate ubiquitination. In patients who retain expression of CUL4A-IpA, although displaying similar IpA usage to PCs

overall, we hypothesize that this truncated protein isoform may provide a mechanism of resistance to lenalidomide. Interestingly, not all cancer cells show depletion of IpA isoforms, as we found increased IpA isoform expression in another B-cell malignancy, chronic lymphocytic leukemia, reported elsewhere.

## Methods

**Samples for 3′-seq and RNA-seq analyses.** Normal B-cell populations derived from tissues were obtained from lymphadenectomies performed at Weill Cornell Medical Center, NY. Blood immune cells were obtained using buffy coats obtained from the New York Blood Center. Mononuclear cells were obtained using Ficoll centrifugation. After that, cells were prepared for fluorescence-activated cell sorting (FACS) to obtain pure populations. Cells were washed with ice-cold phosphate-buffered saline (PBS) once, incubated with appropriate fluorochrome-conjugated antibodies for 30 min at 4 °C and washed twice with ice-cold PBS containing 0.5% fetal calf serum. The following antibodies were used for FACS: anti-CD3-PE (mouse, BD Biosciences, 555333), anti-CD5-FITC (mouse, BD Biosciences, 555352), anti-CD14-PECy7 (mouse, ebioscience, 25-0149-42), anti-CD19-APC (mouse, BD Biosciences, 555415), anti-CD27-PE (mouse BD Biosciences, 555441), anti-CD38-APC (mouse, BD Biosciences, 555462), and anti-CD38-PE (mouse, BD Biosciences, 555460). The purity of immune cell populations was analyzed by FACS and the cells were immediately dissolved in TRI Reagent (Ambion) for RNA extraction, followed by 3′-seq or RNA-seq library preparation. Primary MM cells or PCs were isolated using Ficoll-Hypaque density gradient sedimentation from bone marrow aspirates of MM patients or healthy individuals respectively followed by anti-CD138 microbeads (Milteny Biotech, USA) selection, in accordance with the Declaration of Helsinki following informed consent and Institutional Review Board (Dana-Farber Cancer Institute) approval.

**3′-seq computational analyses.** Preprocessing of 3′-seq libraries, read alignment (hg19)[63], identification, and quantification of peaks were performed as described by Lianoglou et al.[1]. The peaks were assigned to genes using RefSeq annotations. To obtain an atlas of robust cleavage events in 3′UTRs and introns, we started with all the peaks that were detected by peak calling of all the pooled samples and then followed a series of steps to filter lowly expressed peaks and the ones that potentially originate from different artifacts.

Removing artifacts: The peaks potentially resulting from different artifacts were identified and removed: (i) peaks overlapping blacklisted regions of human genome ($n = 2841$; 0.16%) (https://sites.google.com/site/anshulkundaje/projects/blacklists)[64]; (ii) internally primed peaks[1] ($n = 662,562$; 36%); and (iii) antisense peaks ($n = 289,340$; 16%).

Removing the immunoglobin peaks: Our data set included plasma cells which are fully differentiated B cells that secrete antibodies. As plasmas cells produce massive quantities of antibodies, a large fraction of 3′-seq reads mapped to immunoglobulin loci on chromosomes 2 and 14. It was essential to account for this skewed expression of specific genomic regions in order to get a reasonable quantification for the expression of other genes. Thus, peaks ($n = 11$) overlapping with parts of the genome coding for immunoglobulins were removed. Even after this correction, one sample of plasma cells had a high number of intergenic reads (PC2). Thus, this sample was not used for identification of robustly expressed isoforms but only to quantify them. The library size was reduced accordingly for all samples, since the peaks described above result either from sequencing artifacts or from skewed expression of specific genomic regions.

Removing ambiguous peaks: Some genes in the genome overlap with each other. In such cases, it is difficult to assign 3′-seq reads to the genes accurately, and thus such genes ($n = 336$) were removed from further analysis. This resulted in the removal of 8437 peaks from the atlas. Since we were interested in investigating the IpA isoforms of protein coding genes, peaks falling in introns that potentially originated from microRNAs, small nucleolar RNAs, and retrotransposons were also removed ($n = 4722$). Genes that were on the opposite strand but had a 3′UTR end in the intron (100 nt) of a convergent gene can create artifactual antisense peaks in the intron. Thus, peaks in introns that were close to the end of an opposite strand 3′UTR were also removed ($n = 2091$). This corresponded to discarding peaks in the introns of 630 genes. There are genes where the end of the 3′UTR might fall in the intron of the downstream gene on the same strand. This would also create peaks in introns that are contributed by the preceding gene. Therefore, peaks in the intron that were within 5000 nt of the 3′ end of the 3′UTR of the previous gene were discarded ($n = 2079$); the discarded peaks came from the introns of 785 genes.

Identification of robust isoforms: The expression levels of IpA and 3′UTR ApA isoforms were quantified by TPM falling in 3′-seq peaks, i.e., the read count of the peak regions was normalized by the library size of the respective sample. A gene can have many cleavage events with adequate expression levels. To examine cleavage events that represented one of the major isoforms with respect to all isoforms with a 3′ end in a given gene, these isoforms were filtered by usage. Usage is a statistic that gives an estimate of the relative expression of the isoform. As different 3′UTR ApA isoforms create the same protein irrespective of the 3′UTR length, the usage of IpA isoforms was calculated with respect to the total expression of 3′UTR ApA isoforms. IpA isoforms that end in different introns result in

distinct protein isoforms (when translated), and therefore their usage was calculated relative to the total expression of both IpA isoforms and 3′UTR isoforms.

As we were interested in analyzing functionally relevant isoforms, we filtered for robustly expressed isoforms by imposing TPM and usage cutoffs. For the 3′UTR ApA isoforms, an isoform that was expressed with at least 3 TPM and with usage of 0.1 or more became part of the atlas. To focus on the most confident IpA isoforms, an IpA isoform was considered to be robustly expressed only when it was expressed with 5 TPM or more and had 0.1 usage in at least one sample. The interquartile range of the start position of the reads was also required to be 5 or more for the peaks in that particular sample to be defined as a real IpA isoform to eliminate peaks originating from PCR duplicates. These criteria helped to filter the lowly expressed isoforms as well as any possible known artifacts. Filtering for these expression criteria shrunk the atlas from 410,404 peaks to 46,923. As we were interested in IpA and 3′UTR ApA isoforms that would have different functional consequences, peaks that were within 200 nt were clustered to represent a single 3′ cleavage event. Clustering reduced the number of peaks to 40,105. After following the steps above, the atlas comprised 27,927 peaks in 15,670 genes for cleavage events of the 3′UTRs and 3′ ends of 5957 IpA isoforms in 3945 genes. For downstream analysis, we only focused on the IpA isoforms ($n = 5670$) of protein coding genes ($n = 3768$).

**Validation and independent read evidence of IpA isoforms**. We tried to corroborate the robustly expressed IpA events described thus far ($n = 5670$) with external sources of evidence as described below.

1. External annotation: As annotated, we consider mRNA isoforms present in RefSeq, UCSC, or Ensembl. Last exons of all the existing transcripts of the hg19 annotation for Refseq, UCSC, and Ensembl were obtained. These last exons were resized to include a region 100 nt downstream of the annotated end. If the 3′ end of the IpA isoform detected by our 3′-seq analysis overlapped with an expanded last exon, then it was considered to be substantiated by an external annotation. Thus, 39.52% ($n = 2241$) of all the IpA events fell in the vicinity of annotated 3′ end (using the previous definition) based on an external annotation.

2. RNA-seq GLM: RNA-seq read coverage is expected only over the exons and not over the introns, since the splicing machinery splices out introns during co-transcriptional processing of the pre-mRNA. However, if there is an IpA isoform that ends in an intron, then there should be RNA-seq read coverage before the 3′ end of the IpA isoform and no read coverage after the 3′ end (Fig. 1d). To test whether the upstream read coverage was significantly higher than the downstream read coverage, two windows of 100 nt separated by 51 nt upstream and downstream of the IpA 3′ end were defined. These two windows served as replicate bin counts for upstream and downstream coverage. As this was done within each single RNA-seq sample, library size normalization was not required (i.e., the size factor was set as 1 for every comparison). Significant differential expression upstream vs. downstream using DESeq[17] was then tested (FDR-adjusted $P < 0.1$). Not all IpA isoforms could be tested by DESeq. IpA isoforms where the defined windows overlapped with an annotated exon were excluded from further analysis. In total, 4802 events were tested. As a control for this analysis, random introns of expressed genes that did not contain 3′ end peaks were sampled and analyzed as described above. DEseq analysis returned $P$ values consistent with the null hypothesis (Supplementary Fig. 1a). The RNA-seq validation was applied over all the RNA-seq samples. If an IpA event was validated in any sample, then it was considered to be supported by RNA-seq data. Of all the IpA isoforms, 29% ($n = 1670$) could be validated by this approach.

3. Other 3′ end sequencing protocols: If IpA isoforms detected by our 3′-seq protocol were also found by other 3′-end sequencing methods[18], we include the IpA event in our atlas of high-confidence IpA events. This led to the inclusion of 1332 IpA isoforms. The peaks reported by Gruber et al.[18] were resized to be 75 nt width (25 nt upstream of the original start and 50 nt downstream of the original start). Overall, 70% ($n = 3999$) of IpA events were supported by other 3′-end sequencing protocols.

4. Untemplated adenosines from RNA-seq reads (RNA-seq, polyA reads): In RNA-seq data, some reads may overlap the 3′ end of the templated transcript and the start of the polyA tail; these reads contain untemplated adenosines and thus fail to map to the genome. Reads that did not map to the human genome were therefore used to get additional support for the IpA 3′ ends. To make sure that these reads were at the 3′ end, the reads ending with 4 or more As were trimmed. Only reads that were greater than 21 nt in length after trimming were retained. Unmapped reads from all RNA-seq samples were trimmed, and all reads with untemplated As were pooled. These reads were then aligned to the human genome. Using the aligned BAM file, all the reads that were possible PCR duplicates were further filtered out. The uniquely mappable reads that overlapped with the IpA peak (20 nt extended upstream and downstream) were counted. If an IpA isoform was supported by four or more trimmed RNA-seq polyA reads together with the presence of one of the polyadenylation signals (AAUAAA and its variants)[65], then the IpA isoform was considered to be corroborated by polyadenylation RNA-seq reads.

5. Highly expressed IpA isoforms: Since many of our cell types have not been previously assayed by other 3′-end sequencing methods and are also not represented well in existing RNA-seq data sets, we rescued highly expressed cell type-specific IpA isoforms by using a stringent expression cutoff (10 TPM and 0.1

usage). We also required the presence of an upstream polyadenylation signal (AAUAAA and its variants)[65]. This step enabled us to include 323 IpA events in the atlas of highly confident IpA events.

**Expression cutoffs used for IpA and full-length mRNA expression**. A gene is considered to be expressed if either the IpA isoform ($\geq 5$ TPM) or the full-length isoform ($\geq 5.5$ TPM) were expressed in 75% of the samples of the particular cell type.

**Conservation analysis**. We obtained phastCons 46-way conservation scores[21] for 200 nt upstream and downstream of the 3′ ends of IpA isoforms to compare the mean conservation score of the 3′ ends of IpA isoforms against random introns containing polyadenylation signals, but without IpA site usage. The random introns ($n = 5000$) were chosen from IpA genes but we selected introns without IpA events, but with at least one polyadenylation signal (AAUAAA). One of these polyadenylation signals was randomly selected, and we obtained the phastCons 46-way conservation score for 200 nt upstream and downstream of this poly-adenylation signal.

**Identification of differentially used IpA sites**. IpA site usage was calculated as the fraction of reads that map to the IpA site compared to all the reads that map to the 3′UTR of each gene. This translates into the relative expression of the truncated protein compared with the full-length protein. To identify the statistically significant changes in the usage of pA signals, we used a GLM, where we model the read counts of all isoforms across conditions by negative binomial distributions and we test for the significance of an interaction term between isoform and condition. This form of modeling approach was adapted from DEXSeq, which is formulated for testing the differential usage of exons[25]. If a gene has multiple IpA isoforms, then the relative expression of each IpA isoform as well as the pooled full-length mRNA expression were tested independently, since different IpA isoforms are translated into different protein isoforms.

**Gene ontology enrichment analysis**. Functional annotation enrichment was performed on the genes with significant differential usage of IpA sites (Fig. 3a) using DAVID (Database for Annotation, Visualization, and Integrated Discovery) with the expressed genes as the background[26]. Functional annotation enrichment by DAVID was also performed for the genes that loose TMDs and retain TMDs with all the genes that have TMDs expressing IpA isoforms as background.

**Protein domain analysis**. The information about protein domains was obtained from the UCSC UniProt annotation table (spAnnot) via the Bioconductor package-rtracklayer. Only the domains with annotation type 'active site', 'domain', 'trans-membrane region', 'repeat', 'zinc finger region', 'compositionally biased region', 'DNA-binding region', 'region of interest', 'lipid moiety-binding region', 'short sequence motif', 'calcium-binding region', 'nucleotide phosphate-binding region', 'metal ion-binding site', and 'topological domain' from UniProt were used for analysis. These domains were further categorized into more broad categories: (i) Active site—active site and catalytic sites; (ii) DNA-binding domains—C2H2-type, PHD-type, C3H1-type, KRAB, Bromo, Chromo, DNA-binding, C4-type, CHCR, A.T hook, bZIP, bHLH, CCHC-type, CHCH, Bromodomain-like, CH1, C6-type, A.T hook-like, C4H2-type, and CHHC-type; and (iii) Protein–protein interaction domains (PPI)—WD, ANK, TPR, LRR, HEAT, Sushi, EF-hand, ARM, PDZ, PH, SH3, RING-type, LIM zinc-binding, WW, SH2, BTB, FERM, CH, Rod, Coil 1A, MH2, WD40-like repeat, t-SNARE coiled-coil homology, Coil 1B, Cbl-PTB, Coil, CARD, SH2-like, DED, IRS-type PTB, SP-RING-type, EF-hand-like, RING-CH-type, v-SNARE coiled-coil homology, Arm domain, LIM protein-binding, GYF, PDZ domain-binding, and PDZD11-binding. Also, if a region in protein was annotated with 'Interaction with' then that region was considered a PPI domain, (iv) RNA-binding domains—RRM, SAM, KH, DRBM, RBD, Piwi, PAZ, S1 motif, Pumilio, and THUMP; (v) Transmembrane domains (TMDs)—transmembrane region, ABC transmembrane type-1, ABC transporter and ABC transmembrane type-2; and (vi) Repeated—any domains that were repeated in the protein were considered repeated domains. If a gene had multiple protein isoforms, then the longest isoform was used in the analysis. The protein lengths were obtained from http://www.uniprot.org/ for *Homo sapiens*.

**Distance of IpA from TMDs**. IpA isoforms for which there was positional information about the start of first TMD and those that retained at least one domain were used for this analysis. Further, we focused on IpA isoforms that completely lost all their TMDs due to the cleavage event in the intron. The distance of the retained CDR (in amino acids) by IpA from the first TMD was determined as: (upstream CDR from IpA−upstream CDR from the intron before the first TMD)/3.

**De novo transcript assembly**. The complete transcript structure was obtained through the following steps. (i) We used StringTie, an improved method for more accurate de novo assembly of transcripts from RNA-seq data[66]. De novo assembly was performed on every RNA-seq sample with default settings using the hg19

RefSeq annotation. (ii) Transcripts from multiple assemblies were subsequently unified using CuffCompare, which removes redundant transcripts and provides a set of unique transcript structures[67]. (iii) For each individual gene, we obtained the transcripts that overlapped the gene's coordinates. We gave preference to multi-exon transcripts over single exon transcripts. For single exon transcripts, we allowed the start/end to be within 100 nt of the transcription start site (TSS). We gave this advantage to the single exon transcripts because the direction of transcription for these transcripts is not certain. (iv) Finally, using the 3′ ends of IpA isoforms (from our 3′-seq data), we assigned transcripts with the nearest ends to these IpA isoforms.

Firstly, we identified transcripts that ended within 50 nt of 3′-seq events. If there were several assembled transcripts meeting this criterion, we chose the transcript that had the maximum number of exons. If there was a tie in the number of exons, then we chose the transcript that started closest to annotated TSS. For the remaining 3′-seq events, we assigned the nearest ending transcript. Finally, using the above defined criteria for selecting transcript structures, we determined which IpA isoforms corresponded to these assembled transcripts. If the 3′ end of the IpA isoform was within 500 nt of the defined transcript end, then we assumed that this particular transcript represented the full structure of the IpA isoform. For some IpA isoforms we observed usage of different polyadenylation signals within the same intron. Thus, to account for such cases, for the IpA events that did not fall within 500 nt of a transcript end, we determined if it overlapped a transcript that ended within 5000 nt. If this was the case, then we assigned this transcript to that 3′ end. We were able to define the transcript architecture for $n = 954$ IpA isoforms (both annotated and unannotated). If the transcripts ends differed from the IpA 3′-seq events, then we defined the 3′ end determined from 3′-seq to be the real end. This was done as 3′-seq identifies 3′ ends of polyadenylated mRNAs at single nucleotide resolution and thus is more accurate than transcript ends obtained from short read assembly.

**Coding potential prediction**. To determine the probability that the 5′IpA events represented non-coding transcripts, we made use of CPAT, a tool that predicts the coding potential of the transcript based on four sequence features: ORF size, ORF coverage, Fickett TESTCODE statistic, and hexamer usage bias[27]. For our analysis, we considered non-coding IpA isoforms to be the ones that had coding potential probability less than 0.3, had retained coding sequence less than 25%, and had ORF ≤ 300 nt ($n = 469$).

**Binding site enrichment of RNA-binding proteins in exonized introns**. We used available CLIP-sequencing data of RNA-binding proteins from doRiNA[68]. As the majority of CLIP studies were performed in HEK293 cells, we focused on non-coding IpA isoforms expressed in HEK293 and only included IpA isoforms in the analysis whose exonized intron was larger than 50 nt (IpA isoforms = 62, genes = 58).

We determined if the exonized part of the intron was enriched for binding sites of RNA-binding proteins compared to other regions (introns, coding exons, 3′UTRs) of the transcription units, called 'background' here. We calculated the expected number of binding sites in the exonized introns using each background and compared it to the observed number of binding sites in the exonized introns. This enabled us to calculate a binomial Z-score of each CLIP experiment and each background region. We observed enrichment of binding sites of RNA-binding proteins in the exonized introns compared with introns and coding exons but no enrichment compared to 3′UTRs. The RNA-binding proteins with Z-scores ≥10 compared to introns or coding exons are PUM2, FUS, ELAVL1, TIAL1, and TAF15.

**qRT-PCR experiments**. cDNA was synthesized from 200 to 1000 ng of total RNA using a SuperScript VILO cDNA Synthesis Kit (Invitrogen, cat no: 11754050) and random primers. qRT-PCR was performed using SYBR Green PCR Master Mix (Applied Biosystems, cat no: 4309155) on an Applied Biosystems PRISM 7900HT real-time RT-PCR machine using the following primers:

- CUL4A-IpA-F1: TCGTCCCGTTTGTGTCTTCC
- CUL4A- IpA-R1: CCACCTGGACTCCTACGTTC
- CUL4A-full-length-F1: TGGAGAGAGACAAAGACAATCCG
- CUL4A- full-length-R1: TCATGAAGGGGAACCGTCTG
- IQGAP1-IpA-F1: GAGACGTCAGAACGTGGCTTA
- IQGAP1-IpA-R1: AATCTTCTCTCCAGCCAGCC
- IQGAP1-full-length-F1: GACCTAGCCAACAACAGCAC
- IQGAP1-full-length-R1: ACAAATGTCCCATCAGAGCGA
- IKZF1-IpA-F1: TGGGGCTGATGACTTTAGGGA
- IKZF1-IpA-R1: AGTAGCCGCTTGTGTGAGAA
- IKZF1-full-length-F1: TTCCGTGATCCTTTTGAGTGC
- IKZF1- full-length-R1: CTCGCGTTATGTGCGACGA

Data are represented as the ratio of $2^{-\Delta\Delta CTIpA}/(2^{-\Delta\Delta CTIpA} + 2^{-\Delta\Delta CTfull-length})$ for each sample.

**Progression-free survival analysis**. An independent cohort of 319 RNA-seq of MM patients with progression-free survival data for 286 patients was used for this analysis. IpA usage of the isoforms was calculated using RNA-seq data. Expression of IpA isoforms was defined as length normalized reads counts that mapped within 500 bp upstream or $n$ bps (at least 50 bp) without running into an upstream exon from the end 3′-seq peak. Similarly, full-length expression was defined as length normalized reads counts within 500 bp region from the start of the last coding exon of the gene. IpA isoforms that had significantly differential usage (FDR-adjusted $P < 0.05$) in MM when compared to PCs with the difference in usage >0.25 were used ($n = 114$). Only the IpA isoforms that had Pearson's correlation $r > 0.75$ ($n = 28$) were used to define the gene signature. Using this gene signature the patient cohort was defined segregated in two groups with low and high IpA usage, 139 and 176 respectively. Progression-free survival data were available for a subset of these patients in group A ($n = 126$) and group B ($n = 160$). Heterogeneous samples from the two groups were removed based on the group of neighboring samples. For every sample, the nearest five samples were determined based on the Euclidean distance and the sample for which the nearest 80% samples had the same group were retained, group A-filtered ($n = 73$) and group B-filtered ($n = 115$). Progression-free survival data were available for a subset of these patients in group A-filtered ($n = 64$) and group B-filtered ($n = 100$). The R survival package was used to perform the Kaplan–Meier analysis for the two groups.

**Identification of retained introns**. Retained introns were identified using a modified version of the IRFinder algorithm[52]. To avoid genes with a complex genomic architecture, we removed genes that overlap with other genes in either the sense of antisense strand. An intron was categorized as retained if it satisfied the following criteria. (i) There should be at least three reads spanning both the upstream and downstream exon–intron junction. (ii) At least 50% of the intron length should be covered by 3 or more unique reads. Mappability of introns could be a limitation in this case, and thus we focused only on introns that had at least 50% uniquely mappable sequence relative to its complete length. (iii) To ensure adequate expression of the flanking exons, the median coverage over the flanking exons was required to be 10 reads or more. (iv) Since the introns should have more coverage than background noise, we considered introns to be retained if the ratio of median coverage over the intron to median coverage of the upstream and downstream exons was at least 10%.

An intron was annotated as retained if it fulfilled all criteria in at least 66% of the RNA-seq samples of a given cell type. Introns retained in 33% or fewer samples were flagged as not retained while the introns that were retained in more than 33% samples but less than 66% of RNA-seq samples were removed from the analysis. For a 3′ end of an IpA isoform to occur in a particular intron, the intron must contain a polyadenylation signal or one of its variants[65].

Our data showed that some genes had very high coverage over almost all the introns of the gene, presumably due to sequencing artifacts. We determined the (median coverage over all the introns)/(median coverage over all the exons), and if this ratio was ≥0.2 then these genes were flagged for removal.

The number of introns that would have IpA and IR simultaneously by chance were calculated as: probability of IpA × probability of IR × number of expressed introns with polyadenylation signal.

**Data availability**. All 3′-seq and RNA-seq data generated for this study have been deposited in the Gene Expression Omnibus database under accession number GSE111310.

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

## Acknowledgements

This work was funded by NCI grant U01-CA164190 (to C.M. and C.S.L), a grant from the Starr Cancer Foundation (to C.M and C.S.L), the Innovator Award of the Damon Runyon-Rachleff Cancer Foundation and the Island Outreach Foundation (DRR-24-13; to C.M.), the NIH Director's Pioneer Award (DP1-GM123454, to C.M.), the Pershing

Square Sohn Cancer Research Alliance (to C.M.), and the MSK Core Grant (P30 CA008748).

## Author contributions

I.S. designed the computational pipeline for IpA analysis, performed all high-throughput statistical analyses, and carried out integrative analyses with public data sets. S.-H.L. collected the normal B- and T-cell samples and generated all 3′-seq libraries. A.S.S. performed all qRT-PCR validation experiments. I.S. and M.K.S performed the survival analysis. Y.-T.T. and M.F. prepared the MM and PC samples. N.C. M. supervised collection and validation experiments for MM and PC samples and provided clinical interpretation. C.S.L. and C.M. supervised the computational analysis and overall project. C.S.L, I.S., and C.M. wrote the manuscript with input from all authors.

## Additional information

**Competing interests:** The authors declare no competing interests.

