## [Peer Review File · Nature Communications]

Reviewers' comments:

Reviewer #1 (Remarks to the Author):

In this study, the authors generated an atlas for upstream region polyA sites (IPA) using their sequencing method 3'-seq. They examined a number of gene and intron features related to IPA sites. They further analyzed expression changes of IPA isoforms across several cell types in the blood and myeloma cells. Overall, while analysis of IPA in the blood cell context provides some new information for understanding of alternative polyadenylation, the results shown are largely descriptive and represent incremental advancement over previous studies. More in-depth analyses are needed to make this work publishable.

Major:

--The authors need to use their RNA-seq to distinguish skipped terminal exon IPA and composite terminal exon IPA, the two sub-types of IPA. As shown in Tian et al. *Genome Res.* (2007), these two types are quite different with respect to intron size, splice site strength, conservation, etc. The authors have generated RNA-seq data from the same samples, putting them in the perfect position to do such work. Dr. Leslie is an expert on this type of analysis. The result will significantly improve this paper.

Specific comments:

--The high abundance of IPA isoform expression in blood cells (as well as isoforms with short 3'UTRs) was first reported in Zhang...Tian. *Genome Biology*, 2005. This paper should be cited.

--Figure 1b. In the PC1 sample, there are two major peaks on the 3'-seq track, indicating there are two IPA isoforms. Is the first peak real? Can this be a novel isoform or a technical artifact? Can this be detected by other methods, like Northern blot?

--Figure 1e. The authors need to compare with previously cataloged APA events, such those in polyA_DB or by polyA-seq, to show how many novel IPA sites they have identified.

--Figure 2g. The higher conservation of 3'IPA sites may reflect a higher proportion of skipped terminal exon sites in this group, which tend to be more conserved than 5'IPA sites.

--Figures 2h&2i. It is not clear to me if the A[A/U]UAAA and U1 site plots mean anything. This result may be confounded with AT/GC-content of different genes. Large genes are typically in regions with high A/T content, which can explain why IPA genes have a higher A[A/U]UAAA frequency and a lower U1 site frequency (because of several G's) than non IPA genes.

-- Fig 3. It is generally believed that APA events are not switch-like. While addressing this in the Blood NB/NB context may seem new, the result is well expected. It would be more interesting to see if some sequence motifs can be identified around flanking introns/exons which may shed light on IPA regulation mechanism(s).

Reviewer #2 (Remarks to the Author):

Alternative polyadenylation (ApA) generates different mRNA isoforms. The authors of the manuscript have studied ApA (e.g. reference 1), and now are extending the work to characterize

the effect in introns. When present in introns, intronic polyadenylation (IpA) signals can modify the coding regions. This paper generate large list of almost 5,000 IpA sites in humans by analyzing RNA-seq (3' seq and mRNA seq) profiles from a variety of sources (normal, primary immune cells and multiple myeloma (MM)). This paper reports a large number of IpA isoforms in immune cells, that are lost in MM cells.

This is a very nice paper, reporting an interesting phenomenon with a large number of interesting observations. Main concerns are derived from the similarity in analysis from previous work of the team, and the highly descriptive nature of the paper without going into further detail. There are many fascinating observations in the manuscript but they are not followed (I mention a few below). I would suggest the authors to pick up one of the observations and study in detail to reinforce the biological or clinical aspect of the study.

Comments and suggestions:

- the reference to a IpA atlas, in abstract and paper, seems over ambitious given that only a few cell types have been characterized, and that a previous publication from the same group reported more extensive results (in terms of cell types). I suggest to tone down the atlas claim.
- the pipeline and method seems to be the same from previous publication. It would be important to highlight the technical innovation. Leslie's group is a leader in the area and this is an excellent opportunity to emphasize the technical innovation.
- I was wondering why immune cells generate higher number of IpA isoforms. The paper only mentioned "suggesting that the cellular environment has a strong effect on IpA isoform expression", that is a weak statement. Can the authors speculate on molecular mechanisms that operate differentially in these cells? Can they provide some evidence (computational and experimental) in a particular case?
- The comments about the truncated PD-L1 (CD274) need to be further expanded. How often do these truncations occur? Can the authors check in some of the checkpoint inhibitors trial data for expression of the truncated form?
- I am particularly intrigued by the MM analysis associated to figure 6. The authors found that MM cases can be classified in three sets according to IpA site usage. Are these clusters associated to any mutational or expression signature? Is there any correlation with any clinical factor (e.g. prognosis)?
- Another interesting observation is about IKZF1 in MM. It is mentioned, but not followed up in detail. IKZF1 is loss in leukemias (N Engl J Med 2009; 360:470-480). Is this a relevant mechanism there?

Reviewer #1

In this study, the authors generated an atlas for upstream region polyA sites (IPA) using their sequencing method 3'-seq. They examined a number of gene and intron features related to IPA sites. They further analyzed expression changes of IPA isoforms across several cell types in the blood and myeloma cells. Overall, while analysis of IPA in the blood cell context provides some new information for understanding of alternative polyadenylation, the results shown are largely descriptive and represent incremental advancement over previous studies. More in-depth analyses are needed to make this work publishable.

Major:

--The authors need to use their RNA-seq to distinguish skipped terminal exon lPA and composite terminal exon lPA, the two sub-types of lPA. As shown in Tian et al. *Genome Res.* (2007), these two types are quite different with respect to intron size, splice site strength, conservation, etc. The authors have generated RNA-seq data from the same samples, putting them in the perfect position to do such work. Dr. Leslie is an expert on this type of analysis. The result will significantly improve this paper.

We thank the reviewer for suggesting the analysis of the skipped terminal exon vs. composite terminal exon lPA isoforms. Note that we do cite the previous *Genome Research* study, which was published 10 years ago and therefore predates large-scale 3' end and RNA-seq data sets and instead relies on cDNA/EST data as well as older genome annotations. As the reviewer suggests, it is therefore valuable to revisit this idea in light of newer data resources.

If we correctly interpret Figure 1 of the Tian et al. (2007) paper, we understand the distinction between “composite terminal exon” and “skipped terminal exon” as follows. In first case, a donor splice site (5ss) is not recognized and the entire sequence from this donor splice site to the intronic cleavage site is included in the lPA isoform, creating a “composite” last exon; in the second case, the lPA isoform introduces a new small exon ending at the intronic cleavage site, so the previous donor splice site is recognized along with a new acceptor splice site (**Supp. Fig. 2a**).

Indeed, we already performed the needed analysis for examining transcript structure of lPA isoforms through *de novo* assembly of RNA-seq data and comparison to 3'-seq in common cell types. Of the 2675 lPA isoforms for which we could assemble a transcript structure, 1648 (61.6%) displayed loss of recognition of a donor splice site with “composite” terminal exon, while the remainder introduced a new exon (“skipped terminal exon”). We further compared the intron size, transcription unit length, 5ss and 3ss scores for composite vs. skipped terminal exons for these 2675 lPA isoforms (**Supp. Fig. 2b-d**). For splice site scores, we used the maximum entropy model MaxEntScan published by Yeo and Burge, 2004¹.

To summarize our analysis, part of which is confirmatory of the previous cDNA/EST study (**Supp. Fig. 2b-d**), we found:

- 1) Skipped terminal exon lPA isoforms predominantly use 5' lPA sites (i.e. leading to termination in the first 25% of the CDR) while composite terminal exon lPA isoforms use lPA sites throughout the transcription unit.
- 2) Full introns containing the cleavage site for skipped terminal exon lPA isoforms are much longer when compared to composite terminal exon lPA isoforms or introns of genes with no lPA (one-sided Wilcoxon rank-sum test, $P < 10^{-20}$).
- 3) The transcription units of both skipped terminal exon lPA genes and of composite terminal exon lPA genes are longer than those of genes with no lPA (one-sided Wilcoxon rank-sum test, $P < 10^{-20}$; this is a novel result, as transcription unit length was not considered in the previous study).
- 4) The 5ss of composite terminal exon lPA isoforms is weaker than the 5ss of other introns of the same genes (one sided KS-test, $P < 3.35 \times 10^{-7}$).
- 5) The 5ss of skipped terminal exon lPA isoforms is stronger than the 5ss of other introns of the same genes (one sided KS-test, $P < 1.535 \times 10^{-10}$).

- 6) The 3ss of composite terminal lpa isoforms is slightly stronger than the 3ss of other introns of the same genes (one sided KS-test, $P < 1.25 \times 10^{-3}$; the Tian et al. *Genome Research* (2007) study found no significant difference for this case).
- 7) The 3ss immediately upstream of the skipped terminal exon is weaker than the 3ss of other introns of the same genes (one sided KS-test, $P < 1.89 \times 10^{-16}$).
- 8) The 3ss downstream of the skipped terminal exon is stronger than the 3ss of other introns of the same genes (one sided KS-test, $P < 8.43 \times 10^{-10}$).
- 9) 84% of skipped terminal exon lpa events and 80% of composite terminal exon lpa events are associated with AAUAAA/AUUAAA.

Since most of these results revisit earlier findings, we have included the detailed analyses as a supplementary figure (**Supp. Fig. 2a-d**) with a brief summary in the main text, noting which findings are novel relative to the Tian et al. (2007) study.

Specific comments:

--The high abundance of IPA isoform expression in blood cells (as well as isoforms with short 3'UTRs) was first reported in Zhang...Tian. Genome Biology, 2005. This paper should be cited.

We have added a citation to this earlier study and its finding of high expression of lpa isoforms in blood cells, consistent with our 3'-seq analysis.

--Figure 1b. In the PC1 sample, there are two major peaks on the 3'-seq track, indicating there are two IPA isoforms. Is the first peak real? Can this be a novel isoform or a technical artifact? Can this be detected by other methods, like Northern blot?

For this the IGHM locus figure, we originally plotted raw bams as we had removed the peaks of this region to deal with skewness (furthermore, this locus is not annotated in RefSeq and falls in an intergenic region), so they are not part of atlas. It turns out that the first peak, while not dramatically A-rich, is still flagged by our computational pipeline as internally primed. Since the read pile-up is over the coding exon, we cannot appeal to RNA-seq for evidence that it is a real event rather than a technical artifact. For some samples, it seems that there might be a real 3'-end here (pc1, cd38), but in others (pc2) it looks like an artifact where the shape of the peak does not resemble a typical 3'-seq event.

We therefore redid the figure using only regions that are called as peaks in the most raw version of our peak calls – namely, we used only the peaks that are not flagged as internally primed to clean up this track. This removes the first peak.

--Figure 1e. The authors need to compare with previously cataloged APA events, such those in polyA_DB or by polyA-seq, to show how many novel IPA sites they have identified.

As we describe in the paper, one of the sources of evidence we used to assemble an atlas of confident lpa isoforms was consistency with other 3' end sequencing protocols, including PolyA-seq. Briefly, we attempted to validate lpa events from our 3'-seq analysis by sequentially using the following sources of evidence: (1) overlap with the last exon of annotated isoforms in

RefSeq, UCSC and Ensembl; (2) differential upstream vs. downstream RNA-seq coverage; (3) support in data sets from other 3' end sequencing protocols; (4) RNA-seq reads overlapping untemplated adenosines in the polyA tail; (5) high 3'-seq expression in at least one cell type.

Gruber et al. (2016) prepared a Poly(A) site atlas (<http://www.polyasite.unibas.ch/>), which we used to validate our lpA isoforms in step (3) above. Using PolyA-seq data exclusively we can find evidence for 72% of (3581/4927) our atlas isoforms. Meanwhile, using polyA_DB 3 (http://exon.umdj.edu/polya_db/), based on 3'READS data, supports 54% (2677/4927) of our lpA isoforms. One of the main reasons that the other datasets lack evidence for a large fraction of our lpA isoforms is the absence of the immune cell types that we profile in this study.

--Figure 2g. The higher conservation of 3' IPA sites may reflect a higher proportion of skipped terminal exon sites in this group, which tend to be more conserved than 5' IPA sites.

On the contrary, in our analysis we found that skipped terminal exon lpA sites are more predominant in the beginning of the transcription unit, while composite terminal exon lpA isoforms are uniformly present across the transcription unit (**Supp. Fig. 2b**). Therefore, presumably the greater conservation of 3' lpA sites is related to the fact that these isoforms are more likely to generate functional proteins, retaining most of the functional domains of the full-length protein except the most C-terminal domains. The conservations at the lpA site would then reflect evolutionary pressure to maintain expression of these truncated protein isoforms.

--Figures 2h&2i. It is not clear to me if the A[A/U]UAAA and U1 site plots mean anything. This result may be confounded with AT/GC-content of different genes. Large genes are typically in regions with high A/T content, which can explain why IPA genes have a higher A[A/U]UAAA frequency and a lower U1 site frequency (because of several G's) than non IPA genes.

We appreciate the reviewer's caution on this point. To address the issue of the impact of background AT content, we divided both lpA genes and non-lpA genes into those falling in high AT regions versus low AT regions and repeated the pA and U1 signal analysis (**Supp. Fig. 2e**). We again found significant enrichment of pA signals in lpA genes vs. non-lpA genes in both high and low AT regions (one-sided Wilcoxon signed-rank test, lpA high-AT vs non-lpA_high-AT, $P < 10^{-17}$; lpA low-AT vs non-lpA_low-AT, $P < 10^{-16}$) as well as depletion of U1 signals (one-sided Wilcoxon signed-rank test, lpA high-AT vs non-lpA_high-AT, $P < 10^{-15}$; lpA low-AT vs non-lpA_low-AT, $P < 10^{-10}$), although the effect sizes were more modest. However, we further found overall that lpA genes have high AT content compared to the genes that do not contain lpA events, so that AT content is enough to segregate lpA genes from non-lpA genes (**Supp. Fig. 2f**). We now include this caveat in the text to clarify that the difference in AT content is the dominant signal, and that controlling for AT content provides statistically significant but more modest differences.

-- Fig 3. It is generally believed that APA events are not switch-like. While addressing this in the Blood NB/NB context may seem new, the result is well expected. It would be more interesting to see if some sequence motifs can be identified around flanking introns/exons which may shed light on IPA regulation mechanism(s).

While we agree that ApA and lpA events are generally not switch-like, and we show for example that many lpA isoforms are co-expressed across multiple immune cell types, we also report

novel findings about altered IpA expression patterns due to cellular environment and transformation. In particular, we report for the first time that: (1) there is a striking increase in expression of IpA isoforms in blood-derived vs. tissue-derived immune cells of the same lineage, suggesting changes in RNA processing in response to microenvironmental signals; and (2) there is a dramatic loss of plasma cell IpA isoforms in most multiple myeloma (MM) patient samples, including in key genes involved in myeloma biology and response to therapy, such as *IKZF1* and *CUL4A*. We have further added new experimental validation of IpA isoforms at the mRNA level in MM patient samples. Furthermore, guided by an IpA signature defined on MM samples where we have both 3'-seq and RNA-seq data, we performed a large-scale correlative analysis on RNA-seq from 286 MM patients to establish that loss of plasma cell IpA isoforms is significantly associated with shorter progression-free survival (**Fig. 6d**). Further details on these new MM results are described in the response to Reviewer #2.

Early in our analysis for this study, we did look for sequence motifs for RNA-binding proteins that might regulate IpA usage, as suggested by the reviewer, but we did not find clear signals. It is possible that systematic experimental strategies, like screening for RBP factors that alter expression of IpA transcripts, are needed to shed light on IpA regulatory mechanisms, rather than computation-driven approaches. Nevertheless, we believe there are many sources of novelty in the current primarily computational study: we quantify IpA isoforms across tissues and in immune cells and multiple myeloma using new 3'-seq and RNA-seq data sets to characterize IpA genes; we use domain analysis of 3' IpA isoforms to investigate the likely functional changes of truncated protein isoforms; and we present clinically relevant results on the impact of IpA loss in MM, now bolstered with additional validation experiments and correlative studies in a large patient cohort. We believe that all these results represent a major advance in our understanding of the role of IpA in immune cell expression programs.

Reviewer #2

Alternative polyadenylation (ApA) generates different mRNA isoforms. The authors of the manuscript have studied ApA (e.g. reference 1), and now are extending the work to characterize the effect in introns. When present in introns, intronic polyadenylation (IpA) signals can modify the coding regions. This paper generate large list of almost 5,000 IpA sites in humans by analyzing RNA-seq (3'- seq and mRNA seq) profiles from a variety of sources (normal, primary immune cells and multiple myeloma (MM)). This paper reports a large number of IpA isoforms in immune cells, that are lost in MM cells.

This is a very nice paper, reporting an interesting phenomenon with a large number of interesting observations. Main concerns are derived from the similarity in analysis from previous work of the team, and the highly descriptive nature of the paper without going into further detail. There are many fascinating observations in the manuscript but they are not followed (I mention a few below). I would suggest the authors to pick up one of the observations and study in detail to reinforce the biological or clinical aspect of the study.

We thank the reviewer for positive comments about the interesting observations in our manuscript. W.r.t. the similarity in analysis to our previous work, we do acknowledge that we are building on a mature computational pipeline for 3'-seq analysis and for correct statistical assessment of differential 3' end usage. However, to apply our ApA pipeline to detect confident IpA events, we had to implement a series of novel extensions, described in detail in a response

below. We also include novel computational analyses not found in our previous ApA study (Lianoglou et al., 2013), such as: integrative analysis of RNA-seq and 3'-seq, including newly added “composite terminal exon” vs. “skipped terminal exon” analysis requested by Reviewer #1; protein domain analysis to investigate the functional consequences of truncated proteins generated by 3' lpa isoforms; and integration with RBP CLIP-seq data sets.

In order to address the critique that the original manuscript is too descriptive, we have added new experimental and computational results to bolster the clinical significance of our study. In particular, we now show: (1) qPCR validation of lpa isoforms in multiple myeloma (MM) patients and normal plasma cells (PCs), confirming the loss of lpa isoforms of *IKZF1*, *CUL4A*, and *IQGAP1* in a subset of MM patients; (2) new correlative analysis in an independent cohort of 286 MM patients with RNA-seq profiles, demonstrating that the group of patients with greater loss of a signature of PC lpa isoforms has shorter progression-free survival; and (3) new observations, based on literature review and examination of the DDB1-CUL4A-ROC1 crystal structure, to propose the possible impact of IKZF1-lpa and CUL4A-lpa on lenalidomide treatment. We hope that these new results and observations fully address the reviewer's suggestion to reinforce a biological or clinical aspect of our study.

Comments and suggestions:

- the reference to a lpa atlas, in abstract and paper, seems over ambitious given that only a few cell types have been characterized, and that a previous publication from the same group reported more extensive results (in terms of cell types). I suggest to tone down the atlas claim.

We understand the reviewer's concern about overly ambitious language. We use the term “lpa atlas” simply to refer to the set of lpa events defined through systematic computational processing of all the 3'-seq samples in our study; we do not want to claim that the lpa events are fully comprehensive nor that we sample exhaustively across all human tissues. Please note, however, that in addition to the normal B and T lymphocytes, plasma cells, and MM patient samples, we also included all the human tissues and cell lines profiled in our previous ApA study. Moreover, as we show in the paper, the non-immune cell types have fewer lpa events and are biased towards 3' lpa isoforms, supporting our focus on immune cell transcriptomes in order to identify a larger number of novel lpa events (the immune cell focus is also highlighted in the title of the paper). It is fair to say that our lpa atlas is the most comprehensive to date and profiles a range of primary immune cells for the first time by 3'-end sequencing. We have tried to clarify our more restricted use of the term “atlas” as the confident lpa events represented in our samples.

- the pipeline and method seems to be the same from previous publication. It would be important to highlight the technical innovation. Leslie's group is a leader in the area and this is an excellent opportunity to emphasize the technical innovation.

The raw peak detection algorithm is essentially the same as in Lianoglou et al. (2013). However, in order to define an atlas of confident lpa peaks, we added additional steps to ensure that all the potential artifacts are removed:

1. *Removal of black-listed peaks.* Next-generation sequencing based functional genomics experiments (e.g. ChIP-seq, MNase-seq, DNase-seq, FAIRE-seq) tend to produce artificial signal (excessively high read coverage) for certain regions in the genome, which have subsequently been annotated as “blacklisted” regions. These regions have unique mappability

relative to the reference genome and thus are not removed by mappability filters, but they may represent repeat regions whose multiple copies are not reflected in the current genome assembly. In our 3'-seq experiments, we observed very high read mapping for these regions in certain samples. We obtained an exhaustive list of blacklisted regions (<https://sites.google.com/site/anshulkundaje/projects/blacklists>) compiled by the ENCODE consortium (Consortium 2012). 3'-seq peaks in these regions were removed from the atlas (n = 3,926; 0.14%). We also changed the library size for all samples by taking into account the number of reads removed due to blacklisted peaks, as library size is important for sequencing depth normalization.

2. *Removal of peaks associated with PC antibody production.* Our dataset includes plasma cells, fully differentiated B cells that secrete antibodies. As plasma cells express very high levels of antibodies, a large fraction of 3'-seq reads come from parts of the genome (chromosome 2 and chromosome 14) that encode the corresponding Ig polypeptides. It is important to account for this skewed expression of specific genomic regions to accurately quantify expression of other genes. Thus, we deleted the peaks (n = 11) overlapping with parts of the genome coding for antibodies and reduced the library size of all the samples by the number of discarded reads.

3. *Resolution of overlapping opposite-strand 3'UTR peaks.* The genome has genes that fall on opposite strands but have convergent 3'UTRs. Although 3'-seq is a stranded protocol, it can still contain artifactual antisense reads. This poses problems with genes that fall on opposite strands but have 3'UTR ends located within a span of 1000 nt from each other, especially if sense and antisense peaks overlap. To correctly annotate sense and antisense peaks in this situation, we tried to assign the 3'-seq to their respective genes based on the shape and expression of the peaks. For this we made use of an SVM-based supervised learning method that can learn the shape of the sense peaks and distinguish the real sense and antisense peaks for these convergent genes.

4. *Resolution of overlapping opposite-strand 3'UTR and lpa peaks.* The genes that were on opposite strands but have a 3'UTR end in the intron (100 nt) of the convergent gene could create artifactual antisense peaks in the intron. Thus, peaks in introns that were close to the end of an opposite strand 3'UTR were also removed.

5. *Resolution of overlapping same-strand 3'UTR and lpa peaks.* There are genes where the 3'UTR can continue through to an intron of the following gene on the same strand. This can create 3'-seq peaks in introns that are generated by the preceding gene. Therefore, peaks in intronic regions that were within 5000 nt of the 3' end of the 3' UTR of the previous gene were also discarded.

6. *Removal of small RNA-associated peaks.* A lot of microRNAs (miRNAs) and small nucleolar RNAs (snoRNAs) are located in the introns of other genes. We were primarily interested in investigating the lpa isoforms of protein coding genes, and peaks originating from the 3' ends of the miRNAs and snoRNAs located in introns do not represent lpa isoforms. Thus, we decided to remove peaks in introns that were within 500 nt of an annotated miRNA or snoRNA.

7. *Removal of retrotransposon-associated peaks.* To avoid peaks that came from retrotransposons located in the introns, the peaks that overlapped with retrotransposons based on annotations from the ucscRetrolInfo5 track were also removed.

8. *External validation of lpa isoforms.* To assemble an atlas of highly confident lpa events for further analysis, we compared each intronic peak detected in the 3'-seq data against external

annotation and data sources to find additional evidence in support of the lpa isoform (see Methods, Supplementary Fig. 1b, c). Briefly, lpa events that overlapped with the last exon of annotated isoforms in RefSeq, UCSC and Ensembl were first added to the atlas (2,241 events); unannotated lpa events that satisfied the test for differential upstream vs. downstream RNA-seq coverage were added next (907 events); unannotated lpa events without differential RNA-seq coverage but supported in data sets from other 3' end sequencing protocols were then added to the atlas (1,332 events)¹⁸. We then added lpa events that lacked the previous sources of evidence but had RNA-seq support of the cleavage event – i.e. reads overlapping untemplated adenosines in the polyA tail (124 events). Finally, events with high expression in at least one cell type were also included in the atlas (323 events).

These steps are fully described in the Methods section. In addition, with this paper, we are releasing our 3'-seq analysis pipeline as an open source git repository to make the code fully available.

- I was wondering why immune cells generate higher number of lpa isoforms. The paper only mentioned “suggesting that the cellular environment has a strong effect on lpa isoform expression”, that is a weak statement. Can the authors speculate on molecular mechanisms that operate differentially in these cells? Can they provide some evidence (computational and experimental) in a particular case?

We can speculate on the biological reasons why immune cells generate more lpa isoforms than solid tissues. One possibility is that circulating immune cells are specialized for secretion of cytokines, and lpa provides one mechanism for generation of soluble isoforms of proteins whose full-length isoforms are membrane-bound through loss of C-terminal exons encoding transmembrane domains. The canonical example of this phenomenon is the IGHM locus, as discussed in the paper. In addition, the immune system requires repeated execution of complex developmental programs as well as activation-induced differentiation programs. Perhaps immune cells have acquired lpa events to help in the specialization of transcriptional and post-transcriptional programs in cellular differentiation through loss of some (but often not all) repeated DNA-binding and RBP domains. An important case appears to be the transcription factor IKAROS (IKZF1), whose locus predominantly expresses the full-length transcript in mature B cells and predominantly expresses an lpa isoform lacking all DNA-binding zinc finger domains in plasma cells; in a group of multiple myeloma patients, the lpa isoform is lost and the full-length DNA-binding form is upregulated. The activity of IKZF1 is known to be essential for development of B cell precursors², but its DNA binding activity appears to be reduced in plasma cells through a switch to expression of the (apparently non-DNA binding) lpa isoform. Meanwhile, the full-length IKZF1 protein isoform is a key transcription factor in multiple myeloma biology and the target of ubiquitin-mediated degradation through the therapeutic lenalidomide. Therefore, regulation of lpa for a key transcription factor appears to support its differential activity in normal B cell differentiation and in transformation.

We have added a brief summary of these possible explanations for the prevalent use of lpa in immune cells in the **Discussion**.

- The comments about the truncated PD-L1 (CD274) need to be further expanded. How often do these truncations occur? Can the authors check in some of the checkpoint inhibitors trial data for expression of the truncated form?

While we can detect both the lpA and full-length isoforms of CD274 (PD-L1) by 3'-seq in some MM patient samples, TPM read coverage of these events was very low. Moreover, qPCR validation of the lpA isoform in our 15 MM patients was unconvincing (data not shown), again suggesting very low expression of both the lpA and full-length isoforms. We also discussed the potential phenomenon of lpA leading to expression of the soluble form of PD-L1 without the requirement for proteolytic cleavage with Jedd Wolchok's team (Dr. Wolchok leads the immunotherapy/checkpoint inhibitor clinical efforts at MSKCC). It appears that soluble PD-L1 is of unknown clinical significance, as membrane-bound PD-L1 is required for the inhibitory interaction between cancer cells and CD8 T cells, and expression of the truncated isoform is not available in the anti-PD1/anti-PD-L1 trials to which we have access. While circulating soluble PD-L1 has appeared in the biomarker literature for MM³, we were reluctant to pursue further as we could not confirm robust expression of CD274-lpA in MM patient samples, and so instead we focused on other lpA genes (*IKZF1*, *CUL4A*, see below) to strengthen the clinical relevance of our study. We therefore have removed mention of CD274-lpA from our **Discussion**; while this isoform may indeed play a role in other malignancies, the relevance for MM is uncertain.

- I am particularly intrigued by the MM analysis associated to figure 6. The authors found that MM cases can be classified in three sets according to lpA site usage. Are these clusters associated to any mutational or expression signature? Is there any correlation with any clinical factor (e.g. prognosis)?

We are underpowered for this analysis in the current data set as we have only 15 MM patients. However, the Munshi lab has performed gene expression profiling by RNA-seq on a cohort of 286 MM patients with clinical information in a separate study⁴. We used this cohort to determine if the expression of lpA isoforms is correlated with any clinical factor. Before proceeding with the analysis, we needed to establish a set of lpA isoforms whose expression from 3'-seq is robustly represented by RNA-seq signal. We defined the lpA isoform levels in RNA-seq by the expression of the exon upstream of the lpA cleavage site (see **Methods**) and the full-length isoform levels by the last exon of the full-length transcript. The lpA site usage from RNA-seq was then defined as lpA/(lpA + full-length). We computed the correlations of lpA site usage from 3'-seq and RNA-seq across the 12 MM patients for which we had both data types for genes that showed significantly lower usage in MM compared to PCs (FDR-adjusted $P < 0.05$ and difference in usage ≥ 0.25). The distribution of these correlation values (**Supp. Fig. 6a**) shows that RNA-seq signal is often not an accurate readout of lpA isoform expression levels. However, we selected lpA isoforms with high correlation between RNA-seq and 3'-seq (Pearson $r > 0.75$, $N = 28$) to use as a "signature" for the extent of lpA loss in the larger RNA-seq cohort.

Hierarchical clustering w.r.t. the signature lpA events segregated the 286 patients into two groups (see **Methods**, **Supp. Fig. 6b, c**). Group A ($n = 160$) had high median usage of lpA isoforms, while group B ($n = 126$) had low median usage of lpA isoforms. Interestingly, we found that the group A patients, whose lpA expression is more similar to normal plasma cells (PCs), have an improved progression free survival ($P < 0.05$) compared to group B patients (**Supp. Fig. 6d**). We noticed some heterogeneity in the groups, in particular with group B containing some patients with relatively high lpA expression. We therefore applied a k-nearest-neighbor filter (see **Methods**) to remove patients whose neighbors (based on Euclidean distance w.r.t. lpA signature) had inconsistent cluster assignment. This cleaned-up cluster assignment improved the contrast between high lpA group A-filtered ($n = 100$) and low lpA group B-filtered ($n = 64$) and led to a more significant difference in progress-free survival ($P < 0.028$, **Fig. 6d**).

We hypothesize that direct 3'-seq measurement across these 286 patients would lead to a stronger association between clinical outcome and extent of lpA isoform loss. Nevertheless, this analysis establishes a significant correlation between loss of lpA isoforms and shorter progression-free survival, suggesting either that lpA isoform loss is associated with faster progression or that it represents a more advanced disease state compared to MM cells that still express PC lpA isoforms.

- Another interesting observation is about IKZF1 in MM. It is mentioned, but not followed up in detail. IKZF1 is loss in leukemias (N Engl J Med 2009; 360:470-480). Is this a relevant mechanism there?

To bolster the clinical relevance of our study, we validated the relative expression of *IKZF1*, *CUL4A* and *IQGAP1* lpA isoforms at the mRNA level in our cohort of 15 MM patient samples and normal PCs by qPCR (**Fig. 6f**). The ratio of lpA usage by qPCR is very concordant with 3'-seq, confirming the loss of lpA isoforms in a group of MM patients. While we wanted to verify the presence of lpA isoforms for *IKZF1* and *CUL4A* at the protein level, the limited availability of PCs and patient material for western blot analysis prevented us from pursuing this.

We were particularly interested in following up on *CUL4A* and *IKZF1* as *CUL4A* is part of the DDB1-CUL4A-ROC1-CRBN complex that binds lenalidomide and targets *IKZF1* and *IKZF3* for degradation⁵. The CRBN (Cereblon) degron sequence is in a DNA-binding region of *IKZF1* (**Supp. Fig. 7a**), and therefore the lpA isoform that is predominantly expressed in normal PCs cannot be targeted by lenalidomide. We note that *CUL4A* is overexpressed/amplified in other cancers^{6, 7, 8, 9} and that ablation of *CUL4A* has been shown to augment DNA damage¹⁰. Interestingly, a study in prostate cancer cell lines has shown that sensitivity to thalidomide correlates positively with *CUL4A* expression¹¹.

The DDB1-CUL4A-ROC1 crystal structure shows that the N-terminal region of *CUL4A* interacts with DDB1 while the C-terminal region interacts with ROC1, which recruits the E2 ubiquitin-conjugating enzyme (**Supp. Fig. 7b, top**)¹². Furthermore, a truncated *CUL4A* (residues 1-297) has been shown to act in a dominant negative manner¹³. The *CUL4A*-lpA isoform is translated into a truncated protein (1-174 aa) that retains its N-terminal domain and hence the potential to interact with DDB1 (**Supp. Fig. 7b, bottom**). Based on these observations, we hypothesize that *CUL4A*-lpA acts in a dominant negative manner in the group of patients that express the *CUL4A* lpA isoform (group 3 from **Fig. 6a**). In these patients, we predict *CUL4A*-lpA to interact with DDB1, but the complex will not be able to recruit the E2 enzyme required for the substrate ubiquitination. In the **Discussion**, we hypothesize that although this group of patients shows similar lpA usage to PCs, they may potentially display resistance to thalidomide treatment through this mechanism.

We have now added (1) the qPCR validation of lpA isoforms in MM patients and PC samples; (2) the correlative analysis showing association between lpA loss and shorter progression-free survival; as well as (3) discussion of the loss of the Cereblon degron sequence in the *IKZF1*-lpA protein isoform and the potential role of the *CUL4A*-lpA protein isoform as a dominant negative. These results and observations have clear implications for the efficacy of lenalidomide treatment in MM and may suggest a mechanism for resistance through *CUL4A*-lpA expression. While we are planning to pursue more detailed functional validation experiments to prove this hypothesis, such experiments may take more than the few months we had for preparing the revisions and are perhaps out of scope for this paper. However, taken together, we believe

these new results and observations demonstrate the clinical importance of IpA isoform expression.

Finally, to address the reviewer's question about the B-ALL study in NEJM, it is clear from the genetics of B-ALL that IKZF1 acts there as a tumor suppressor, while its role in MM is as an oncogene: "Deletion of *IKZF1*, which encodes the lymphoid transcription factor IKAROS, is a very frequent event in *BCR-ABL1*-positive ALL and at the progression of chronic myeloid leukemia to lymphoid blast crisis". So in this setting, we might expect an increased usage of the short isoform, rather than exclusive expression of the full-length isoform. Indeed, in a separate study on another hematopoietic malignancy, chronic lymphocytic leukemia (CLL), we report increased expression of IpA isoforms rather than loss of IpA (Lee*, Singh* et al., in review)¹⁴. However, since we do not have 3'-seq samples in B-ALL, and since increased IpA in cancer cells is the subject of our other paper, we do not further address this hypothesis here.

References

1. Yeo G, Burge CB. Maximum entropy modeling of short sequence motifs with applications to RNA splicing signals. *J Comput Biol* **11**, 377-394 (2004).
2. Georgopoulos K, Winandy S, Avitahl N. The role of the Ikaros gene in lymphocyte development and homeostasis. *Annu Rev Immunol* **15**, 155-176 (1997).
3. Wang L, et al. Serum levels of soluble programmed death ligand 1 predict treatment response and progression free survival in multiple myeloma. *Oncotarget* **6**, 41228-41236 (2015).
4. Cleyne A, et al. Expressed fusion gene landscape and its impact in multiple myeloma. *Nat Commun* **8**, 1893 (2017).
5. Kronke J, et al. Lenalidomide causes selective degradation of IKZF1 and IKZF3 in multiple myeloma cells. *Science* **343**, 301-305 (2014).
6. Melchor L, et al. Comprehensive characterization of the DNA amplification at 13q34 in human breast cancer reveals TFDP1 and CUL4A as likely candidate target genes. *Breast Cancer Res* **11**, R86 (2009).
7. Wang Y, et al. CUL4A induces epithelial-mesenchymal transition and promotes cancer metastasis by regulating ZEB1 expression. *Cancer Res* **74**, 520-531 (2014).
8. Hung MS, et al. Cul4A is an oncogene in malignant pleural mesothelioma. *J Cell Mol Med* **15**, 350-358 (2011).
9. Yang Y, Wang S, Li J, Qi S, Zhang D. CUL4A as a marker and potential therapeutic target in multiple myeloma. *Tumour Biol* **39**, 1010428317703923 (2017).
10. Liu L, et al. CUL4A abrogation augments DNA damage response and protection against skin carcinogenesis. *Mol Cell* **34**, 451-460 (2009).

11. Ren S, *et al.* Oncogenic CUL4A determines the response to thalidomide treatment in prostate cancer. *J Mol Med (Berl)* **90**, 1121-1132 (2012).
12. Angers S, Li T, Yi X, MacCoss MJ, Moon RT, Zheng N. Molecular architecture and assembly of the DDB1-CUL4A ubiquitin ligase machinery. *Nature* **443**, 590-593 (2006).
13. Chen X, *et al.* A kinase-independent function of c-Abl in promoting proteolytic destruction of damaged DNA binding proteins. *Mol Cell* **22**, 489-499 (2006).
14. Lee S SI, Tisdale S, Abdel-Wahab O, Leslie C and Mayr C. Widespread intronic polyadenylation inactivates tumor suppressor genes in leukemia. (ed^(eds) (2017).

REVIEWERS' COMMENTS:

Reviewer #1 (Remarks to the Author):

The authors have addressed all my concerns.

Reviewer #2 (Remarks to the Author):

The authors fully addressed my comments.